# Identification of Antisense RNA NRAS-AS and Its Preliminary Exploration of the Anticancer Regulatory Mechanism

**DOI:** 10.3390/genes15121524

**Published:** 2024-11-27

**Authors:** Liping Wang, Xuming Hu, Chenyue Tao, Jacob Xiang, Hengmi Cui

**Affiliations:** 1Department of Biobank, Northern Jiangsu People’s Hospital Affiliated to Yangzhou University, Yangzhou 225001, China; njphwlp@163.com; 2College of Animal Science and Technology, Institute of Epigenetics and Epigenomics, Yangzhou University, Yangzhou 225001, China; hxm@yzu.edu.cn; 3Jiangsu Key Laboratory for Animal Genetic, Breeding and Molecular Design, College of Animal Science and Technology, Yangzhou University, Yangzhou 225001, China; 4School of Nursing School of Public Health, Yangzhou University, Yangzhou 225001, China; taoqiac@163.com; 5Clinical Pharmacist, Foothills Medical Centre, 140329 St NW, Calgary, AB T2N 2T9, Canada; jzxiang@ualberta.ca

**Keywords:** hepatocellular carcinoma, antisense RNA, NRAS-AS, DNA methylation, AZA

## Abstract

Objective: To explore the influence of NRAS-AS on the proliferation, apoptosis, cell cycle, migration, and invasion ability of HCC cells, as well as its underlying mechanisms. Methods: A double-stranded cDNA library for liver cancer cells was constructed, and identified NRAS-AS through High-throughput sequencing, bioinformatics, chain-specific fluorescent quantitative PCR, and RACE. NRAS-AS′s effects on HepG2 and SMMC-7721 cells and gene expression were evaluated. Additionally, the study analyzed the influence of NRAS-AS overexpression on tumor formation in nude mice. Immunohistochemistry and Western blotting were used to detect NRAS protein levels in clinical samples. RT-qPCR examined NRAS-AS and NRAS gene expression in HCC and adjacent tissues. Results: NRAS-AS overexpression suppresses HCC cell proliferation and invasion, induces cell cycle alterations in HepG2 and SMMC-7721 cells, and enhances apoptosis. NRAS-AS interference promoted liver cancer invasion, inhibited apoptosis, and influences the cell cycle. Nude mice overexpressing NRAS-AS showed smaller tumors. NRAS-AS expression in liver cancer patients correlated with clinical factors. RT-qPCR revealed an inverse correlation between NRAS-AS and NRAS gene expression in liver cancer and adjacent tissues. IHC analysis revealed reduced NRAS protein expression in HepG2 and SMMC-7721 cells following NRAS-AS overexpression. The impact of AZA treatment on antisense NRAS-AS and sense NRAS gene expression in liver cancer cells was observed, and antisense. Conclusion: Reduced NRAS-AS expression is frequently observed in HCC and is inversely related to NRAS gene expression, suggesting a role in HCC pathogenesis through NRAS regulation. Targeting antisense RNA NRAS-AS could hold promise as a therapeutic target and diagnostic biomarker for HCC.

## 1. Introduction

Hepatocellular carcinoma (HCC) is a common malignant tumor of the digestive system. Its occurrence and development are complex processes involving multiple factors, steps, and stages. According to the latest statistics, HCC ranks second in the mortality rate of tumor-related diseases in China [1]. HCC is characterized by its insidious onset, high malignancy, rapid progression, and poor prognosis, posing a serious threat to the lives and health of the Chinese population and even the global population. It has become an urgent and significant public health issue in need of a solution.

Antisense RNA is a widely existing endogenous regulatory RNA in humans and mice [1,2]. It is also known as natural antisense transcripts (NATs). By forming double-stranded RNA (dsRNA) with sense transcripts (STs), it regulates the levels of mRNA and plays an important role in regulation [3]. Many NATs, or antisense RNA, are classified as long non-coding RNA (lncRNA) [4]. The study by Gao et al. discovered that the lncRNA HOTAIR is aberrantly expressed in tumors [5]. Interaction with the PRC2 or LSD1 protein complex induces the demethylation of histone H3K4me2 in tumor-associated genes at the gene level, thereby promoting malignant proliferation, suppressing apoptosis, and facilitating metastasis of tumor cells. Overall, antisense RNA regulates important events such as the growth, apoptosis, and migration of tumor cells by affecting the expression of sense genes [6,7]. However, little is known about the role and molecular mechanisms of antisense RNA in the occurrence and development of tumors.

In recent years, the important role of activation of oncogenes and inactivation of tumor suppressor genes in the process of tumor development has gradually been revealed. Among them, the inactivation of tumor suppressor genes mainly occurs through three pathways: gene abnormal methylation changes, gene deletion, and gene mutation. Modern tumor theory suggests that at the genetic level, the formation of tumors can be explained by genetic and epigenetic mechanisms. The oncogene Neuroblastoma ras sarcoma homolog (NRAS) is located on human chromosome 1 and is an important member of the ras sarcoma (RAS) family of proto-oncogenes. It belongs to the small GTPase protein family. NRAS gene mutations are common in various tumor diseases, such as multiple myeloma, subcutaneous melanoma, colorectal cancer, and adult acute myeloid leukemia (AML), and play an important role in the occurrence and development of these tumors [8,9].

Research has reported the occurrence of NRAS mutations in adult patients with acute myeloid leukemia (AML). According to the FAB (France, American, Britain) classification, these mutations are mainly observed in the M5 subtype, and patients have shown improved survival rates after initial induction therapy [10]. Chen et al. found that upregulating NRAS expression by relieving its inhibition activates the MAPK signaling pathway, promoting melanoma cell proliferation, inhibiting apoptosis, and facilitating cell migration and invasion [11]. In colorectal cancer, the mutation rate of KRAS reaches 40–50%, while the mutation rate of NRAS is approximately 5–9%. Studies have found that tumors with NRAS mutations exhibit resistance to EGFR treatment [12]. Therefore, NRAS can regulate the growth, migration, invasion, and other activities of tumor cells through multiple pathways, including the activation of PI3K/AKT and NF-κB signaling pathways [13]. Although the role of NRAS has been extensively studied in various malignant tumor diseases, research on its role in the occurrence and development of HCC is relatively scarce.

It is known that epigenetic modifications play a crucial role in the occurrence and development of HCC. As a key epigenetic regulatory method, DNA methylation often affects the expression of numerous genes, including some oncogenes and tumor suppressor genes that are closely related to tumorigenesis. 5-aza-2′-deoxycytidine (AZA) is a classic and effective DNA methylation inhibitor. It can reverse the abnormal hypermethylated state of DNA by inhibiting the activity of DNA methyltransferases, thus potentially enabling genes that were previously silenced due to hypermethylation to resume expression. We speculate that in HCC, there may be some regulatory RNAs related to tumor progression that are under the control of methylation and are in a state of low expression or have not yet been discovered. Therefore, we introduced AZA to try to explore whether it could induce the expression of potentially important functional RNAs and then uncover their mechanism of action in HCC. Secondly, many previous studies have reported that in research on other types of tumors, after treatment with AZA, some new non-coding RNAs with the function of regulating the biological behaviors of tumor cells were discovered. These successful precedents suggest to us that in HCC research, we can also utilize AZA as a powerful tool to search for key RNA molecules that may be involved in the functional regulation of HCC cells.

This study selected HCC, one of the main malignant tumors in China, as the research subject. By ingeniously using DNA methylation inhibitor AZA to treat HCC cells and constructing a double-stranded cDNA library, along with utilizing methods such as genome-wide high-throughput sequencing, the antisense RNA NRAS-AS was identified for the first time. Preliminary studies have found that DNA methylation inhibitors inhibit the activity of DNA methyltransferase, alleviate the inhibition based on methylation, reverse abnormal DNA hypermethylation, affect the expression of NRAS-AS, and then affect the expression of NRAS and inhibit the occurrence and development of HCC. This discovery holds promise for the early diagnosis, treatment, and prognosis evaluation of HCC and has significant implications for both basic research and clinical medical research.

## 2. Materials and Methods

### 2.1. Patients and Samples

A total of 45 HCC patients who underwent surgical resection in the Hepatobiliary Surgery Department of the Chinese PLA General Hospital from February 2018 to July 2020 were selected for this study. Among them, there were 28 males and 17 females, ranging in age from 49 to 80 years old, with tumor diameters ranging from 2 to 20 cm. All subjects included in the study had not received radiotherapy, chemotherapy, or other anti-tumor treatment before surgery. Tissue samples were collected in a manner that did not affect clinical diagnosis, and both adjacent non-cancerous and cancerous tissues were collected and stored for future use. The survival information of the matched HCC paraffin samples was recorded for 5 years. All samples were histologically diagnosed by two pathologists. The collection of samples has been reviewed by the Ethics Committee of the Chinese PLA General Hospital, and informed consent forms have been signed.

### 2.2. Cell Lines

The HCC cell lines HepG2 (RRID: CVCL_0027) and SMMC-7721 (RRID: CVCL_0534) were obtained from the cell repository of the Chinese Academy of Sciences (Shanghai), while the HEK293 cell line (RRID: CVCL_0045) was obtained from the American Type Culture Collection (ATCC). The aforementioned cells were cultured in DMEM medium (Gibco, Grand Island, NY, USA) supplemented with 10% fetal bovine serum FBS (Gibco) and 1% penicillin/streptomycin (Gibco) and incubated in a constant temperature cell culture incubator at 37 °C and 5% CO_2_.

### 2.3. Establishment of HepG2 Cell Double-Stranded cDNA Library

HepG2 cells were treated with AZA, and RNA was extracted. This was followed by reverse transcription to synthesize cDNA and further processes, including DNA nucleotide kinase S1 treatment, resulting in the construction of a double-stranded cDNA library for HepG2 cells. The control group was treated with DMSO. This method can be used for the identification of novel antisense RNA in liver cancer. The flowchart is shown in Figure 1.

### 2.4. Establishment of a Double-Stranded cDNA Library for HepG2 Liver Cancer Cells and High-Throughput Genomic Sequencing Analysis

HepG2 cell was treated with a DNA methylation inhibitor, AZA (at a concentration of 5 μM), and Dimethyl Sulfoxide (DMSO) for 48 h. Total RNA was extracted from the HepG2 cell after treatment with DNase I to remove genomic DNA. Subsequently, cDNA was synthesized through reverse transcription, and RNase H was used to remove RNA from the samples. The samples were incubated overnight at 55 °C for self-hybridization of complementary sequences. Then, Nuclease S1 was used to remove all single-stranded DNA, generating double-stranded cDNA from HepG2 cells. The double-stranded cDNA libraries of the AZA treatment group and the DMSO control group HepG2 cells were subjected to next-generation genomic sequencing analysis by Suzhou Genewiz Intelligent Company, with three biological replicates for each group. Sequencing results were processed and optimized using software such as CASAVA, Trimmomatic, and Fast QC to identify image base recognition, optimize data, and evaluate quality. From these results, liver cancer-related genes with carcinogenic effects were selected, and the presence of sense and antisense RNA in their sequences was determined. Finally, gene functional annotation and enrichment pathway analysis were performed using tools such as Gene Ontology (GO) and Pathway analysis.

### 2.5. Identification of the Full-Length Sequence of NRAS-AS Using RACE Technology

The rapid amplification of cDNA ends (RACE) technique was employed to identify the full-length sequence of NRAS-AS. The SMARTer^®^ RACE 5′/3′Kit from Takara was used to perform the RACE experiment. Firstly, total RNA was extracted using TRIzol^®^ Reagent, Using the SMART Scribe Reverse Transcriptase, 1μg of genomic DNA-free RNA was reverse transcribed to generate 5′ or 3′ RACE products. The amplification was then carried out using the universal primer UPM and gene-specific primers (5′-GSP or 3′-GSP primers), and the antisense transcript end sequence was obtained through cloning and sequencing. Each RACE experiment was performed in triplicate, and the sequencing was done for at least three independent clones to ensure the reliability of the results.

### 2.6. Vector Construct

The lentiviral vector expressing NRAS-AS was constructed, and primers containing NehI and BamHI restriction sites were designed. The pcDNA3.1-NRAS-AS plasmid was used as a template to PCR-amplify the full-length sequence of NRAS-AS. The PCR product was then ligated into the lentiviral vector pCDH-CMV-MCS-EF1-Puro after restriction digestion to construct the pCDH-NRAS-AS vector. Subsequently, lentiviral packaging was performed in HEK-293T cells using a four-plasmid co-transfection system to generate the pLV-NRAS-AS lentivirus. The virus solution was used to infect HepG2 and SMMC-7721 cells, and stable expression cell lines were selected with puromycin. The NRAS-AS monoclonal cell line was selected by the dilution method, and its DNA, RNA, and protein were extracted and analyzed. Finally, PCR and Sanger sequencing were used to verify the integration and expression of NRAS-AS in the monoclonal cell line.

### 2.7. RT-qPCR 

Total RNA from all tissues and cells was extracted using Trizol reagent, and cDNA was synthesized with the Vazyme gDNA removal reverse transcription kit. The strand-specific primers for NRAS and NRAS-AS, as well as the primers for real-time fluorescence quantitative PCR, were designed and synthesized. The strand-specific primers were diluted to a 25-fold concentration, and the RT-qPCR primers were diluted to a 10-fold concentration. The RT-qPCR reaction system was prepared using Vazyme SYBR Premix Ex Taq II, which contained 10 μL of SYBR Green Master Mix, 1 μL of forward primer (10 μM), 1 μL of reverse primer (10 μM), and 1 μL of cDNA template, with the final volume adjusted to 20 μL. Amplification was carried out according to the reaction program of pre-denaturation at 95 °C for 30 s, followed by 35 cycles, each cycle consisting of denaturation at 95 °C for 5 s, annealing, and extension at 60 °C for 30 s. Subsequently, the cycle threshold method (ΔΔCt) was adopted to analyze the RNA expression levels, with GAPDH and U6 as the normalization reference standards. After PCR amplification, the melting curve data were analyzed using the BIO-RAD CFX Connect^TM^ real-time PCR system. Each sample was subjected to three biological replicates to ensure the reliability of the results. 

### 2.8. Cell Proliferation

The cells were inoculated into 96-well plates; each well was added with 10 μL CCK-8 solution (Vazyme) and incubated at 37 °C for 2 h. The light absorption value at 450 nm was measured for six consecutive days, and the growth curve was drawn. In addition, both groups of cells were evenly seeded into a 6-well culture plate until clones appeared, at which point the culture was immediately stopped. The cells fixed with 4% neutral formaldehyde were stained with crystal violet solution for 15 min, photographed, and the number of clone formations was calculated. 

### 2.9. Flow Cytometry

Logarithmically growing cells were resuspended in 100 µL of 1× Binding Buffer. Then, 5 µL of Annexin V-FITC and 5 µL of PI Staining Solution (Vazyme) were added, and the mixture was incubated in the dark for 10 to 15 min. Then, 400 µL of 1× Binding Buffer was added and gently mixed. Cell apoptosis was detected using a flow cytometer with an excitation wavelength of 488 nm. In addition, cells were fixed with pre-chilled 70% ethanol for 2 h. Then, 500 µL of PI staining solution (Vazyme, Nanjing, China) was added to the tested cell samples. After thorough mixing, the samples were incubated in the dark at 37 °C for 30 min. Cell cycle analysis was performed using a flow cytometer, and subsequently, Modfit analysis software was used to analyze cell DNA content and light scattering.

### 2.10. Cell Migration Experiment

Cells were seeded into pre-marked six-well plates, and when the cells reached approximately 95% confluence, a cell scratch was made. 1.5 mL of serum-free DMEM medium was added to each well, and the distance between the cells on both sides of the scratch was observed at different time points, such as 0 h, 24 h, 48 h, and 72 h, to compare the migration of NRAS-AS. In addition, 100 μL of cell suspension with a concentration of 10 × 10^6^ cells/mL was slowly added to the upper chamber coated with Matrigel Matrix, and 600 μL of complete medium containing 10% FBS was added to the lower chamber. After 48 h of incubation, the upper chamber of the transwell was removed, and the cells were fixed with anhydrous methanol. Crystal violet staining solution was added, and the cells were stained for 20 min at room temperature. The cells were observed and photographed under a microscope, and the number of invaded cells was counted.

### 2.11. Animal Experiment

Twelve four-week-old BALB/C-Nu nude mice were purchased from the Experimental Animal Center of Yangzhou University and housed in an SPF-level animal facility. The experiment was divided into a control group and an experimental group, each consisting of six mice. Both groups were inoculated with a cell concentration of 2 × 10^7^ cells/mL. In the control group, HepG2-GFP was subcutaneously injected into the right forelimb axilla of nude mice, while in the experimental group, HepG2-NRAS-AS was injected. The injection volume for each mouse was 200 μL. The nude mice in both the control and experimental groups were observed every other day for the occurrence of tumors. After tumor formation, the size of the tumor was measured, and the volume was calculated for further analysis. Within four weeks after inoculation, the nude mice were sacrificed, and subcutaneous tumor tissues were collected. The tumor tissues were then fixed with formalin, embedded in paraffin, and used for immunohistochemical staining. The remaining half of the tumor tissue was used for protein extraction and subjected to Western blot analysis.

### 2.12. Immunohistochemical

The tumor tissues of the two groups were sequentially fixed, dehydrated, embedded, sliced, antigen repair, and antigen-binding (1:500 Anti-NRAS antibodies were added overnight at 4 °C), followed by DAB staining and neutral glue sealing, etc. Finally, the images were photographed under a microscope, and the comprehensive optical density (IOD) of the images was analyzed by IPP software 6.0.

### 2.13. Western Blotting

The tumor tissue protein was extracted, and its concentration was determined. After gel electrophoresis, membrane transfer, sealing, adding 1:500 diluent of NRAS primary antibody, incubation at 4 °C overnight, then adding 1:500 sealing solution to dilute HRP labeled secondary antibody, incubation at room temperature for 1 h. Then, the quantitative analysis was carried out by using Image J software 1.8.0 through chemiluminescence visualization and other methods.

### 2.14. Statistical Analysis

Data were analyzed using SPSS 26.0 software. In this study, *t*-tests and analysis of variance (ANOVA) were used for intergroup comparisons of quantitative data. The chi-square test was used for intergroup comparisons of qualitative data to analyze the correlation between gene expression and clinical pathological characteristics. The prognosis and survival of HCC patients were analyzed using methods such as Kaplan-Meier and log-rank tests. *p* < 0.05 was considered statistically significant.

## 3. Results

### 3.1. Collection of Clinical Data Related to HCC Patients’ Specimens

In this study, standardized and complete clinical data from 45 liver cancer patients were collected. The collected data include patients’ names, gender, age, history of chronic hepatitis, degree of cirrhosis, levels of AFP and total bilirubin, tumor size, staging, and differentiation level. Follow-up information was obtained from the clinical resources and biological sample follow-up database, with the follow-up period starting from the date of the patients’ surgery and ending on 31 March 2022. The clinical characteristics data of HCC patients are shown in Table 1.

### 3.2. Screening and Verification of Antisense RNA of the Oncogene NRAS

By constructing a double-stranded cDNA library of HepG2 cell line and performing high-throughput genome sequencing and bioinformatics analysis, we found a total of 545 dsRNAs of liver cancer oncogenes in the DNA methyltransferase inhibitor-treated group and DMSO-treated group (Figure 2A). Based on the preliminary screening results, we selected *JUN*, *SET*, and *NRAS* as the next research targets. First, we synthesized cDNA using specific primers for sense *NRAS* and antisense NRAS-AS chain. Before performing conventional PCR amplification, we treated the samples with Dnase I. The results showed that cDNA synthesized using the specific primers for sense NRAS and antisense NRAS-AS chain could both produce distinct amplification bands. In addition, we set up a control group without adding chain-specific reverse transcriptase (RT-) and did not observe any amplification bands after PCR amplification, which excluded the possibility of genomic DNA contamination. This indicates the successful construction of a double-stranded cDNA library of the HepG2 cell line and the first identification of antisense RNA of *JUN*, *SET* and *NRAS* genes (Figure 2B). In combination with recent literature reports, this study focuses on the exploration of the oncogenes NRAS and NRAS-AS.

The full-length sequence of NRAS-AS was observed by agarose gel electrophoresis. Both 5’RACE PCR and 3’RACE PCR products showed an obvious band, and the band size was consistent with the theoretical value in the database (Figure 2C). 

### 3.3. The Effects of NRAS-AS Overexpression on the Biological Functions of HCC Cells

To investigate the effects of NRAS-AS on the biological functions of liver cancer cells, we successfully constructed an NRAS-AS overexpression vector. CCK-8 and colony formation assays showed that overexpression of NRAS-AS effectively inhibited the viability of HCC cells (Figure 3A–C). Flow cytometry analysis revealed that overexpression of NRAS-AS promoted apoptosis in liver cancer cells (Figure 3D,E). In addition, a decrease in the proportion of cells in the G0/G1 phase and an increase in the proportion of cells in the S and G2/M phases were observed. This indicates that the antisense RNA NRAS-AS plays a regulatory role in the cell cycle of HepG2 cells. Wound healing (Figure 3F,G) and transwell (Figure 3H,I) assays demonstrated that overexpression of NRAS-AS led to a decrease in the invasive ability of HCC cells in both HepG2 and SMMC-7721 cell lines and migration ability was significantly reduced. Compared with the empty vector control group, the proportion of HepG2 cells overexpressing NRAS-AS in the G0/G1 phase decreased, while the proportions of cells arrested in the S phase and G2/M phase increased, indicating that the DNA replication of the cells was significantly more active (Figure 3J,K). These results further support the investigation of NRAS-AS as a potential therapeutic target for anti-tumor treatment.

### 3.4. The Effect of Knockdown of NRAS-AS on the Biological Function of HCC Cells

To further investigate the influence of NRAS-AS on the biological functions of HCC cells, we knocked down NRAS-AS and assessed the knockdown efficiency. Based on the results, ASO-NRAS-AS-1 was chosen for the subsequent experiments (Figure 4A). Flow cytometry analysis revealed a statistically significant decrease in early apoptosis rate in cells with NRAS-AS knockdown (*p* < 0.05) (Figure 4B,C). Additionally, we observed an increase in the proportion of HepG2 cells in the G0/G1 phase and a decrease in the proportion in the S phase, from 44.2% to 10.7% (Figure 4D,E). This suggests that NRAS-AS knockdown may induce cell cycle arrest from the G0/G1 to S phase. Transwell assay results showed a significant increase in the number of HCC cells successfully invading the extracellular matrix within 24 h (*p* < 0.01) (Figure 4F,G). This indicates an enhanced invasive ability of liver cancer cells after NRAS-AS knockdown. These results further support the potential of NRAS-AS as a therapeutic target for anti-tumor treatments.

### 3.5. The Effect of NRAS-AS Overexpression in Nude Mice and HCC Cells

We constructed a subcutaneous tumor model in nude mice and plotted the tumor volume growth curve. The results showed that after overexpression of NRAS-AS, the tumor volume formed by HepG2 cells in nude mice was significantly smaller and lighter (Figure 5A–C). IHC of tumor tissues revealed a significant decrease and even loss of NRAS protein expression (Figure 5D). Western blotting results also showed that in the NRAS-AS overexpression group, the expression of NRAS protein was significantly lower compared to the control group (Figure 5E,F). These preliminary results confirmed the inhibitory effect of NRAS-AS on the tumorigenic ability of HepG2 cells in vivo, demonstrating strong anti-tumor potential. This suggests that NRAS-AS may suppress the tumorigenic ability of HCC cells by regulating the expression of NRAS protein. Additionally, the expression of NRAS-positive protein in HCC cells was significantly reduced after overexpression of NRAS-AS (Figure 5G). 

It is speculated that the expression of NRAS protein may be regulated by NRAS-AS, showing a downward trend. Furthermore, treatment of HepG2 cells with AZA revealed a significant upregulation of NRAS-AS expression (*p* < 0.01) and a downward trend in NRAS expression (*p* < 0.05) after AZA treatment (Figure 5J). These preliminary results provide the basis that DNA methylation inhibitors can inhibit the activity of DNA methyltransferase, alleviate the methylation-based inhibition, reverse the abnormal DNA hypermethylation, and then affect the expression of NRAS and regulate the down-regulation of oncogenes. This study provides a basis for further research on the mechanisms of antisense RNA regulation of oncogenes.

### 3.6. Correlation Analysis Between NRAS-AS Expression Level and Clinical Pathological Features in HCC Patients Tissues

In order to study the relationship between NRAS-AS expression level and clinical pathological parameters, this study divided liver cancer tissues into a high-expression group and an expression group according to the median relative expression level of NRAS-AS. The results showed that there was a significant correlation between high NRAS-AS expression in liver cancer patients and TNM staging of tumor tissue (*p* = 0.011), capsule condition (*p* = 0.013), tumor size (*p* = 0.008), total bilirubin level (*p* = 0.003), and AFP level (*p* = 0.023) (*p* < 0.01), while there was no correlation with patient’s gender (*p* = 0.299), age (*p* = 0.302), and differentiation degree (*p* = 0.427), and there was no significant statistical difference (*p* > 0.05). Table 2.

### 3.7. Expression Patterns of NRAS-AS and NRAS Genes in Clinical HCC Patient Tissue Samples

IHC results showed that in well-differentiated and moderately differentiated liver cancer tissues, *NRAS* staining was significantly higher than in the adjacent non-cancerous tissues, with a brown-yellow and brownish distribution of granules, indicating that the expression level of *NRAS* protein in HCC tissues was higher than in the corresponding adjacent non-cancerous liver tissues (Figure 5H). RT-qPCR results revealed that in the cancer tissues and adjacent non-cancerous tissues of 45 HCC patients, the expression level of NRAS-AS was lower in the cancer tissues compared to the adjacent non-cancerous tissues, while the expression level of *NRAS* was higher in the cancer tissues. The expression level of NRAS-AS showed a negative correlation with the expression of the *NRAS* (Figure 5I). 

### 3.8. Prognostic Analysis of NRAS-AS Expression in HCC Patients’ Survival

Furthermore, we divided NRAS-AS into a high-expression group and a low-expression group and analyzed the relationship between NRAS-AS expression and the survival of HCC patients using Kaplan-Meier analysis (Figure 6A). The results showed that HCC patients in the high-expression group of NRAS-AS had significantly higher five-year survival rates than those in the low-expression group of NRAS-AS (*p* = 0.023), with a statistically significant difference. The survival rate of female patients was significantly lower than that of male patients (*p* = 0.0079) (Figure 6B), the survival rate of patients with total bilirubin level > 17.1 μmol/L was significantly lower than that of patients with ≤17.1 μmol/L (*p* = 0.0051) (Figure 6D), the survival rate of patients with AFP ≥ 25 μg/L was significantly lower than that of patients with AFP < 25 μg/L (*p* = 0.0019) (Figure 6E), and the lower the tissue differentiation degree, the worse the survival rate (*p* = 0.0012) (Figure 6F), while there were no significant differences in age (*p* = 0.26) (Figure 6C), capsule (Figure 6G), tumor stage (Figure 6H), tumor size (*p* = 0.36) (Figure 6I) and hepatitis (Figure 6J), etc.

## 4. Discussion

Although it is known that genetic mutations are associated with tumor formation, increasing evidence suggests that epigenetic variations are also closely related to the occurrence, development, and metastasis of tumors [14]. Current research data shows that the molecular mechanisms of epigenetic silencing of tumor suppressor genes have been widely revealed [15]. However, regarding the mechanistic role of oncogenes in the abnormal activation of tumor cells, apart from reports related to genetic mutations, no relevant reports have been found at the level of epigenetic mechanisms. The discovery of epigenetic silencing of tumor suppressor genes has introduced epigenetic drugs as a method for clinical treatment. For example, AZA and its analog 5-Azacytidine, as inhibitors of DNA methylation, have been used in clinical research for the treatment of leukemia and certain solid tumors [16,17]. With the rapid development of high-throughput sequencing technology, research on antisense RNA and its role in tumor occurrence, development, and treatment has attracted widespread attention in the medical field. In this study, the oncogene NRAS was chosen as an effective target gene, hoping to explore its potential value in inhibiting tumor growth, excessive proliferation, and antisense RNA therapy by studying the inhibition or knockout of abnormally expressed NRAS oncogene in tumor cells.

In this study, the AZA demethylation regulation and induction of antisense RNA expression were used to construct a double-stranded cDNA library of HepG2 cells for high-throughput genome sequencing. Simultaneously, combining bioinformatics and chain-specific fluorescent quantitative PCR technology, the presence of the NRAS-AS gene was screened and identified for the first time. After obtaining the sequencing sequence of NRAS, the complete NRAS-AS cDNA of the 5′ and 3′ ends was amplified from liver cancer cells using RACE technology, with a total length of 804 bp. Compared with the homology reported in GenBank for liver cancer *NRAS*, the homology reached over 99.88%. Therefore, we named it NRAS-AS. The acquisition of the full-length sequence of NRAS-AS provides an important basis for further study of its biological functions.

In this study, overexpression of NRAS-AS was found to inhibit the proliferation, invasion, and migration capabilities of HCC cells and promote apoptosis. Research has shown that NRAS plays an important role in cell proliferation and migration by binding with GTP/GDP and GTPase. Under normal physiological conditions, *NRAS* can also control cell proliferation and migration by activating the RAS/RAF/MAPK and PIK3CA/AKT signaling pathways downstream of the Epidermal Growth Factor Receptor (EGFR), which in turn can lead to carcinogenesis. Transwell and wound healing assays showed that overexpression of NRAS-AS in HepG2 and SMMC-7721 cells resulted in decreased invasiveness and significantly reduced migration capabilities of HCC cells, consistent with findings in breast cancer studies. Conversely, interference with NRAS-AS increased the invasiveness of HCC cells. This study also found that NRAS-AS plays a regulatory role in the cell cycle of HepG2 cells. The use of nude mouse tumor models, which can stably reflect the biological and genetic characteristics of primary tumors, has been widely used in tumor treatment and related research, offering advantages such as high tumor formation rates and good uniformity [18,19].

We constructed a nude mouse subcutaneous tumor model and plotted the tumor volume growth curve. The results showed that the subcutaneous tumors formed by overexpressing NRAS-AS in HepG2 cells were significantly smaller and lighter in mass. Immunohistochemical staining and Western blotting of the tumor tissues also showed a significant decrease or even absence of NRAS protein expression in the NRAS-AS overexpression group. Similar trends were observed in the cells as well. These results preliminarily confirmed the inhibitory effect of NRAS-AS on the tumorigenic ability of HepG2 cells in vivo, demonstrating its strong anti-tumor potential. In vitro, NRAS-AS was found to participate in various physiological activities such as cell proliferation, migration, and apoptosis, inhibiting the growth process of HepG2 cells. It is predicted that NRAS-AS may regulate the occurrence and development of tumors by controlling the expression of *NRAS* protein, laying the foundation for further research into the specific molecular regulatory mechanisms.

To evaluate the relationship between the expression levels of NRAS-AS and the occurrence of HCC, this study first analyzed the correlation between NRAS-AS and the clinical characteristics of HCC patients, suggesting that NRAS-AS plays an important role in the pathological staging of liver cancer. The survival analysis results showed that patients with high expression of NRAS-AS had a significantly higher five-year postoperative survival rate than those with low expression of NRAS-AS (*p* = 0.023). The survival rate of female patients was significantly lower than that of male patients (*p* = 0.0079), and the survival rate of patients with total bilirubin level > 17.1μmol/L was significantly lower than that of patients with ≤17.1 μmol/L (*p* = 0.0051). The survival rate of patients with AFP ≥ 25 μg/L was significantly lower than that of patients with AFP < 25 μg/L (*p* = 0.0019). Additionally, the lower the tissue differentiation degree, the worse the survival rate (*p* = 0.0012), while there was no difference with age (*p* = 0.26) or tumor size (*p* = 0.36). RT-qPCR results showed a negative correlation between the expression of NRAS-AS and NRAS (Figure 5I). Histologically, the expression level of NRAS protein in HCC tissues was higher than that in adjacent tissues. We collected HepG2 cells treated with AZA and found that the expression level of NRAS-AS was significantly upregulated (*p* < 0.01), while the expression of *NRAS* showed a downward trend (*p* < 0.05) (Figure 6). Based on the above results, DNA methyltransferase inhibitors can inhibit the activity of DNA methyltransferase, alleviate the inhibition based on methylation, reverse abnormal DNA hypermethylation, affect the expression of NRAS-AS, and then affect the expression of *NRAS* and inhibit the occurrence and development of HCC. It is speculated that AZA can affect the expression of oncogene *NRAS* by affecting the antisense RNA NRAS-AS, thereby inhibiting the development of liver cancer (Figure 7).

*NRAS* and its antisense RNA NRAS-AS play a key role in tumor development and may affect the immune response of HCC by regulating the expression of *NRAS* protein. NRAS-AS not only changes the biological behavior of tumor cells, such as the expression of immune molecules, and interferes with the immune escape mechanism but also may affect the function and activity of immune cells in the tumor microenvironment, including the polarization direction of macrophages and the activity of NK cells, thereby breaking the immune balance. In addition, NRAS-AS may also interfere with the immune surveillance process, allowing tumors to evade recognition by the immune system and further break the immune regulatory balance. Although current studies focus on the direct effects of NRAS-AS on HCC cells, its immunoregulatory role deserves further exploration, which may provide a new perspective for HCC treatment and immune strategies.

Although this study systematically analyzed the effects of NRAS-AS on the behavior and mechanisms of HCC cells, there are some limitations. First, the limited clinical sample size may miss potential associations or introduce bias. Therefore, larger multicenter studies are needed to verify the value of NRAS-AS in the diagnosis, prognosis, and treatment of HCC. Furthermore, in vitro and in vivo models have limitations in representing human disease. Results from cell lines and mouse models alone do not always translate to clinical outcomes in HCC patients, especially given the heterogeneity of human cancers. In conclusion, despite the valuable findings, more comprehensive studies are needed to elucidate the role of NRAS-AS in HCC and its potential as a therapeutic target and diagnostic marker.

In fact, this study also demonstrated the significant role of NRAS-AS in the proliferation, apoptosis, and invasion of liver cancer cells in in vitro experiments. However, due to the relatively small number of clinical samples included in this study, with only 45 cases, the experimental results have a certain degree of singularity and limitations. In the future, it is necessary to expand the sample size and further explore the expression of NRAS-AS in liver cancer, as well as the relationship between NRAS-AS and the regulation of *NRAS* protein expression in the occurrence and development of liver cancer, in order to elucidate the anti-cancer molecular regulatory mechanisms of antisense RNA NRAS-AS in the process of liver cancer occurrence and development.

## 5. Conclusions

In this study, a DNA demethylating agent (AZA) was used to treat liver cancer cells to induce the expression of antisense RNA. Through high-throughput genomic sequencing screening and identification, the full-length sequence of the oncogene antisense RNA NRAS-AS was successfully obtained. We preliminarily confirmed that overexpression of NRAS-AS could significantly inhibit the proliferation, migration, and invasion ability of liver cancer cells in vitro and in vivo while promoting apoptosis and affecting cell cycle distribution. Furthermore, in the experiment of interfering with NRAS-AS in vitro, it was observed that the proliferation, migration, and invasion abilities of cells were promoted, while apoptosis of liver cancer cells was inhibited, and it affected cell cycle distribution. We also found that differential expression of NRAS-AS was related to clinical characteristics, verifying the expression patterns of NRAS-AS and *NRAS* genes in liver cancer tissues and adjacent tissues. This is the first discovery that AZA affects the expression of the oncogene *NRAS* by activating the expression of NRAS-AS, thus inhibiting the occurrence and development of liver cancer and revealing this molecular regulatory mechanism. Therefore, antisense RNA NRAS-AS is a potential drug for HCC treatment and a potential biomarker for the diagnosis of hepatocellular carcinoma.

## Figures and Tables

**Figure 1 genes-15-01524-f001:**
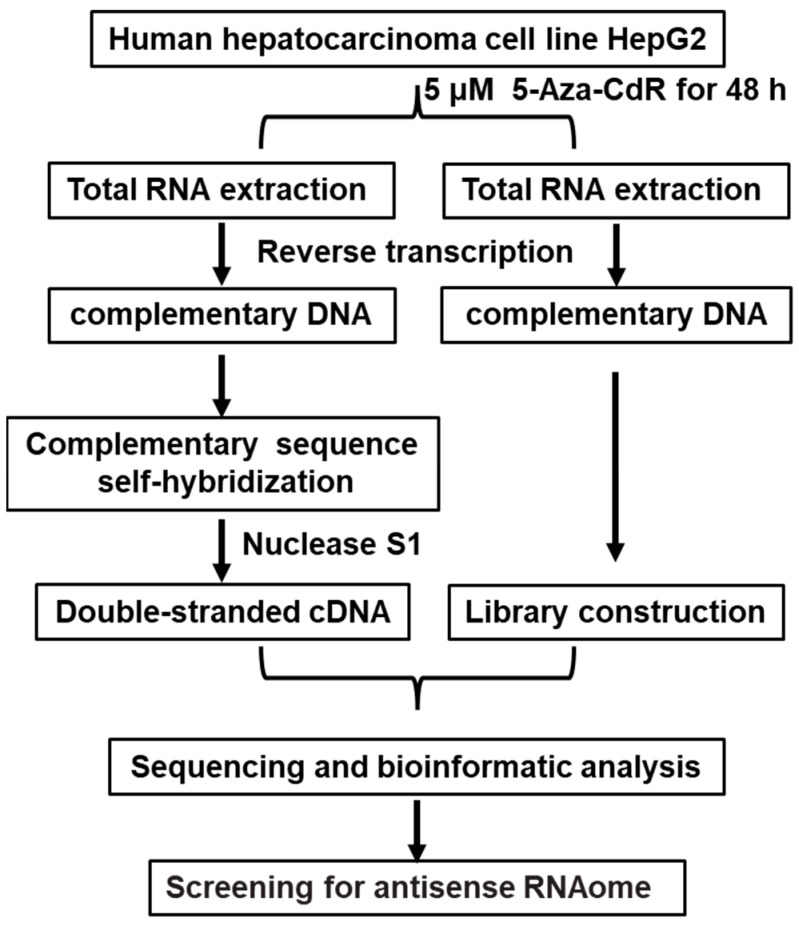
Construction of double-stranded cDNA of HepG2 cells.

**Figure 2 genes-15-01524-f002:**
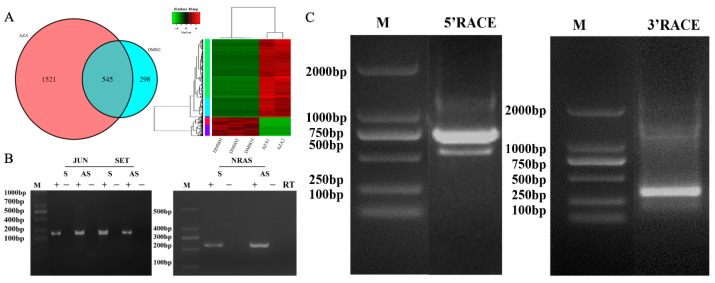
Screening and verification of antisense RNA of the oncogene NRAS. (**A**): intersection analysis and heat map analysis of sequencing data between the AZA treatment group and the DMSO control group; (**B**) The verification of sense and antisense RNA in JUN, SET, NRAS gene; (**C**) The agarose gel electrophoresis results of NRAS-AS 5′RACE and 3′RACE PCR products. Note: Marker: 500 bp, 2000 bp; S, AS, and RT stand for strand RNA, antisense strand RNA, and reverse transcription, respectively.

**Figure 3 genes-15-01524-f003:**
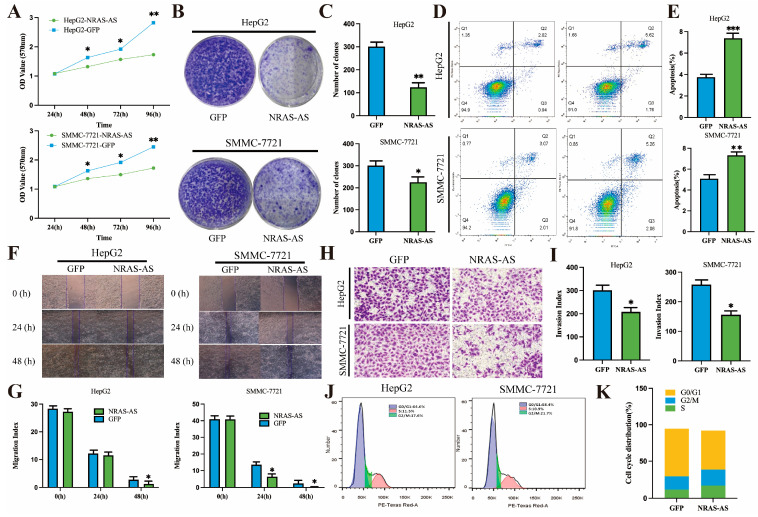
Effect of overexpression of NRAS-AS on the biological function of HCC cells. (**A**) The effect of NRAS-AS on the proliferation of HCC cells was detected by CCK8 assay; (**B**,**C**) Effects of NRAS-AS on the proliferation of HCC cells were detected by Colony formation assays; (**D**,**E**) Overexpression of NRAS-AS promotes apoptosis in HCC cells; (**F**,**G**) Overexpression of NRAS-AS inhibits migration in HCC cells; (**H**,**I**) Overexpression of NRAS-AS inhibited the invasion in HCC cells; (**J**,**K**) Effect of NRAS-AS overexpression on cell cycle in HepG2 cells. * *p* < 0.05, ** *p* < 0.01, *** *p* < 0.001, n = 3.

**Figure 4 genes-15-01524-f004:**
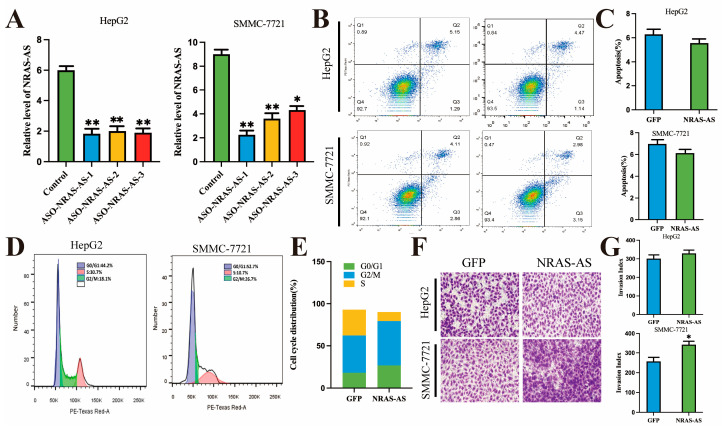
Effect of Knockdown of NRAS-AS on the biological function of HCC cells. (**A**) Knockdown efficiency of NRAS-AS detected by RT-qPCR; (**B**,**C**) Knockdown of NRAS-AS inhibits apoptosis in HCC cells; (**D**,**E**) Effect of knockdown NRAS-AS on cell cycle in HepG2; (**F**,**G**) Knockdown of NRAS-AS promoted the invasion ability in HCC cells. ** *p* < 0.01, * *p* < 0.05, n = 3.

**Figure 5 genes-15-01524-f005:**
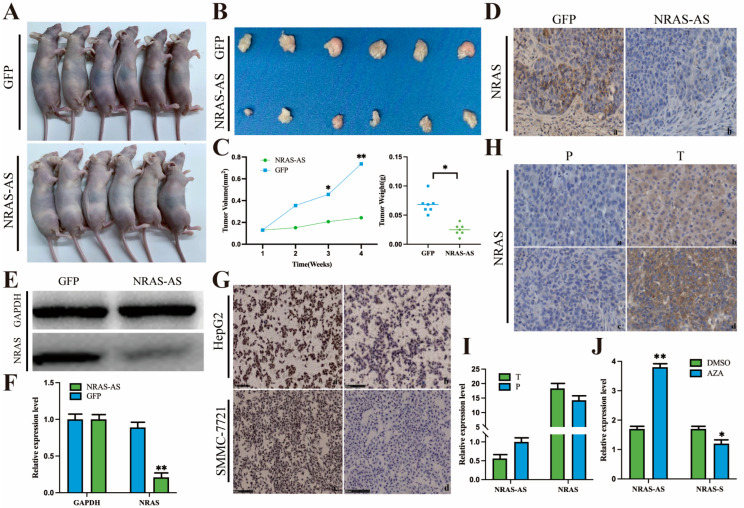
Effect of *NRAS-AS* overexpression in nude mice. (**A**) The nude mice of two groups; (**B**,**C**) The nude mice tumor mass and volume of two groups; (**D**) Expression of NRAS in tumor tissues of nude mice overexpressing NRAS-AS was determined by IHC (×400); (**a**) The expression of GAPDH in the tissues of GFP and NRAS-AS nude mice; (**b**) Expression of NRAS in the tissues of GFP and NRAS-AS nude mice; (**E**,**F**) Expression of NRAS in tumor tissues of nude mice overexpressing NRAS-AS was determined by WB; (**G**) The expression of NRAS in overexpressed NRAS-AS HCC cells detected by IHC; (**a**) The IHC results of NRAS in HepG2-GFP cell; (**b**) The IHC results of NRAS in HepG2-NRAS-AS cell; (**c**) The IHC results of NRAS in SMMC-7721-GFP cell; (**d**) The IHC results of NRAS in SMMC-7721-NRAS-AS cell; (**H**) Expression of NRAS in HCC carcinoma and adjacent tissues (×400), (**a**) The IHC results of NRAS in the adjacent tissues of well-differentiated HCC; (**b**) The IHC result of NRAS in well-differentiated HCC; (**c**) The IHC result of NRAS in adjacent tissues of moderately differentiated HCC; (**d**) The IHC results of NRAS in moderately differentiated HCC; (**I**) Expression of NRAS-AS and NRAS in HCC tissues; (**J**) Effect of AZA on differential expression of NRAS-AS and NRAS in HCC cells. ** *p* < 0.01, * *p* < 0.05.

**Figure 6 genes-15-01524-f006:**
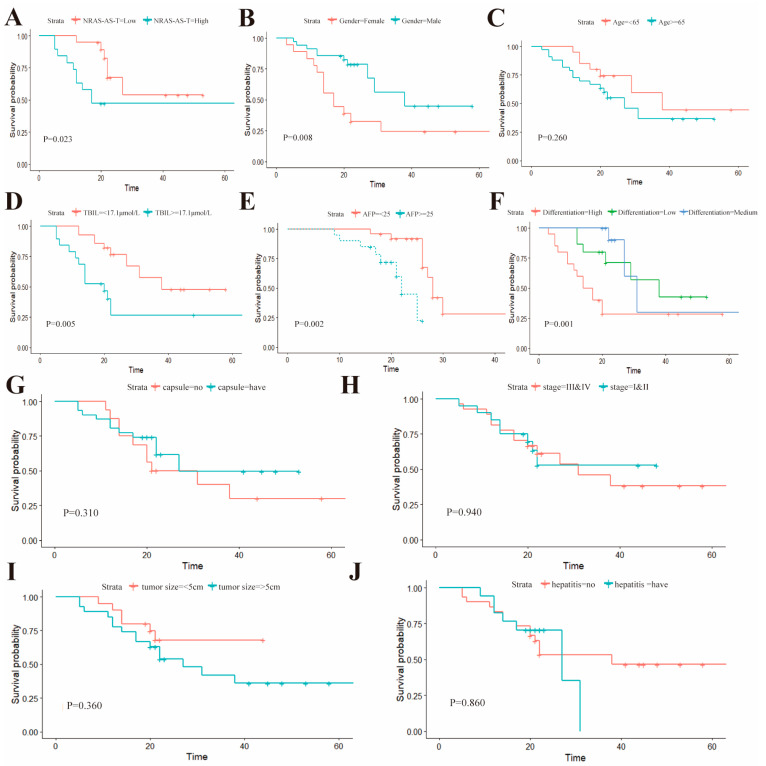
NRAS-AS expression and prognosis of patients undergoing HCC resection. (**A**) Survival analyses were compared between low and high NRAS-AS-T groups; (**B**) Survival analysis comparing female and male patients; (**C**) Comparison of survival analyses in patients <65 years and ≥65 years; (**D**) TBIL levels and survival analysis were compared between the TBIL < 17 μmol/L group and the TBIL ≥ 17 μmol/L group; (**E**) AFP levels and survival analysis were compared between AFP < 25 and AFP ≥ 25 groups; (**F**) Survival analysis of the highly differentiated and the moderately poorly differentiated groups; (**G**) Survival analysis in capsule and non-capsule groups; (**H**) Stage I/II and Stage III/IV survival analysis; (**I**) Survival analysis of tumor size <5 cm and ≥5 cm; (**J**) Survival analysis of groups with and without hepatitis.

**Figure 7 genes-15-01524-f007:**
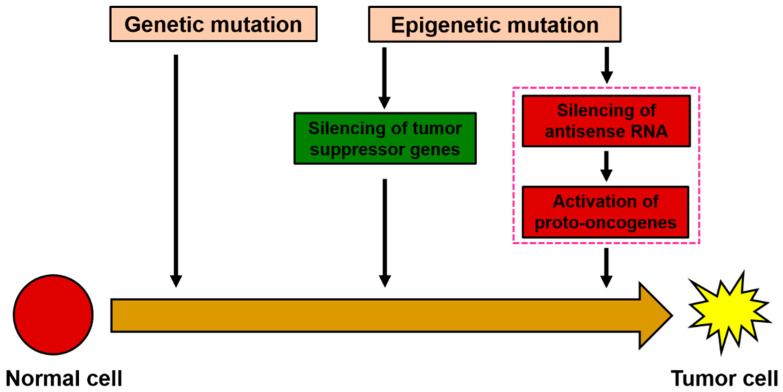
The mechanism hypothesis diagram.

**Table 1 genes-15-01524-t001:** Clinical characteristics of HCC patients.

Clinical Pathological Characteristics	Categories	Quantity	Sum %
Gender	Female	17	37.78
	Male	28	62.22
Age	Age < 65	19	42.22
	Age ≥ 65	26	57.78
Total bilirubin	Total bilirubin < 17.1	18	40.00
	Total bilirubin ≥ 17.1	27	60.00
AFP	AFP ≥ 25	17	37.78
	AFP < 25	28	62.22
Histodifferentiation	Moderate differentiation	25	55.56
	Low differentiation	13	28.89
	High differentiation	7	15.56
Envelope	Not	14	31.11
	Have	31	68.89
Clinical stages	I–II	21	46.67
	III–IV	24	53.33
Tumor size	Tumor size < 5	17	37.78
	Tumor size ≥ 5	28	62.22
History of chronic hepatitis	Not	26	57.78
	Have	19	42.22
Liver cirrhosis	Not	8	17.78
	Have	37	82.22
Sum		45	

**Table 2 genes-15-01524-t002:** Correlation analysis between NRAS-AS expression levels and clinical pathological characteristics.

Clinical Characteristics	Categories	NRAS-AS Level		Sum	χ²	*p*
		Low Expression	High Expression			
Gender	Female	7(41.18)	10(58.82)	17	1.079	0.299
	Male	16(57.14)	12(42.86)	28		
Age	Age < 65	8(42.11)	11(57.89)	19	1.067	0.302
	Age ≥ 65	15(57.69)	11(42.31)	26		
Total bilirubin	Total bilirubin < 17.1	14(77.78)	4(22.22)	18	8.538	0.003 **
	Total bilirubin ≥ 17.1	9(33.33)	18(66.67)	27		
AFP	AFP ≥ 25	5(29.41)	12(78.59)	17	5.148	0.023 *
	AFP < 25	18(64.29)	10(35.71)	28		
Histodifferentiation	Moderate differentiation	11(44.00)	14(56.00)	25	1.701	0.427
	Low differentiation	7(53.85)	6(46.15)	13		
	High differentiation	5(71.43)	2(28.57)	7		
Envelope	Not	11(78.57)	3(21.43)	14	6.133	0.013 *
	Have	12(38.71)	19(61.29)	31		
Clinical stages	I–II	15(71.43)	6(28.57)	21	6.505	0.011 *
	III–IV	8(33.33)	16(66.67)	24		
Tumor size	Tumor size < 5	13(76.47)	4(23.53)	17	7.032	0.008 **
	Tumor size ≥ 5	10(35.71)	18(64.29)	28		
History of chronic hepatitis	Not	15(57.69)	11(42.31)	26	1.067	0.302
	Have	8(42.11)	11(57.89)	19		
Liver cirrhosis	Not	6(75.00)	2(25.00)	8	2.222	0.136
	Have	17(45.95)	20(54.05)	37		
Sum		23(51.11)	22(48.89)	45		

* *p* < 0.05 ** *p* < 0.01.

## Data Availability

All data generated or analysed during this study are included in this article and its Appendix A.

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
