# Peer review of "Identification of Antisense RNA NRAS-AS and Its Preliminary Exploration of the Anticancer Regulatory Mechanism"

_genes, 2024, doi:10.3390/genes15121524_

Round 1
Reviewer 1 Report
Comments and Suggestions for Authors
This is a nice scientific contribution. The methods are chosen and performed appropriately, the results sound interesting, and the manuscript is well-written. Of note is that the methods consist of various bioinformatics, high-throughput assays, molecular genetic tests, wet-lab biology methods, animal experiments, and clinical data analysis.
Some points:
1. Line 88: Please recheck the number of male and female patients.
2. Lines 113-119: Please review the text for any repeated information.
3. Line 120: In which section of the study is "image-based recognition" conducted?
4. Section 2.5. RT-qPCR: Please provide the primers' sequences and the reaction program's details.
5. Lines 147-153: Please review the text for any repeated information.
6. Please provide some details of the NRAS immunohistochemistry and western blotting methods.
7. It looks like Table 1 is not uploaded. I could not find it in the submitted material.
8. Figure 1 and its associated text belong to the Materials and Methods section.
9. Line 255: Please provide details on how you constructed the NRAS-AS vector.
10. Figure 5E: please recheck the labels. It looks like NRAS and GAPDH are written in place of each other.
11. Line 331: “Correlation analysis of NRAS-AS expression level and clinical pathological features” looks redundant.
12. Please include mean (+/-SD) survival time for NRAS-AS and NRAS groups.
13. Line 380: “The presence of the NRAS gene was screened and identified for the first time”. Did you mean the NRAS-AS gene?
14. Line 427: Please recheck the reference to Fig. 6 in this sentence. The RT-qPCR information is shown in Figure 5.
15. Figure 7. This is a nice diagram, but I wonder why the authors believe that epigenetic alterations occur in later stages of tumorigenesis, as they are drawn after “genetic mutations”.
Author Response
Our specific responses have been uploaded to the file.

Reviewer 2 Report
Comments and Suggestions for Authors
The authors of the present work studied the effect of NRAS-AS on HCC cell lines. The results showed a reduction in proliferation, migration apoptosis and invasion related to the NRAS-AS expression.
The manuscript looks like well written and organized. The authors have presented an interesting topic in the field of oncology and cancer treatments. The paper should be considered after major revisions.
1. The title does not describe properly the subject of the article. The results are too preliminary to suggest an anticancer effect of the antisense RNA. Moreover, its effect was observed only in HCC patient’s outcomes and only two cell lines were used to perform in vitro and in vivo experiments. The authors should change the title;
2. The authors report the regulatory effects of NRAS-AS on many cell mechanisms, including apoptosis and proliferation. The decreasing of viability probably induces the decreasing of the migratory and invasiveness capabilities and to the cell cycle alterations. In this scenario, the data related to the alteration of cell mechanisms become redundant. What is the author’s point of view on this aspect? ;
3. The Figure 2 is not clear. In particular, Fig 2D is not informative and Fig2E is difficult to understand and read. The authors should move the Fig2D in the supplementary data and modify the Fig 2E;
4. In many results paragraphs, it is not mention the figure numbers. The authors should report the figure number associated to the relative data description;
5. The study is based on the identification of new therapeutic targets for the treatment of HCC patients. Well-defined preclinical models are needed to better understand the behavior and drug response of cancer cells. Commercial cell lines cultured on common monolayer supports are in vitro systems that are not able to mimic the microenvironment of cancer diseases. On the other hands, in vivo models represent some valuable research resources to reproduce the TME. For these reasons, the authors should underline these aspects through a short overview of preclinical models developed to reproduce the TME and to underline the importance to integrate different systems in order to increase the complexity of the tool used. The following references should be included in the manuscript: “doi: 10.1038/s41416-023-02186-1”, “doi: 10.3390/jpm12060854” and “doi: 10.1186/s41199-020-00056-4”;
6. Study limitations should be added.
Author Response
- The title does not describe properly the subject of the article. The results are too preliminary to suggest an anticancer effect of the antisense RNA. Moreover, its effect was observed only in HCC patient’s outcomes and only two cell lines were used to perform in vitro and in vivo experiments. The authors should change the title;
Response 1: Thank you very much for your very pertinent suggestions.
Your point that the title does not appropriately describe the theme of the article really hits the nail on the head. As you said, our current research results are still at a preliminary stage. Although we have carefully designed and carried out relevant in vivo and in vitro functional experiments respectively to explore the mechanism of action of NRAS-AS in HCC, from the perspective of the overall depth and breadth of the research, the existing evidence is indeed insufficient to strongly prove that NRAS-AS has a definite anti-HCC effect. Moreover, we also realize that at present, relevant phenomena have only been observed in the prognosis of HCC patients, and in terms of experiments, only two cell lines have been used for in vitro and in vivo experiments, which indeed has certain deficiencies in rigor and comprehensiveness. Therefore, taking into comprehensive consideration your valuable opinions and the actual situation of our research at the current stage, after careful discussion by our research team, we have unanimously decided to change the article title to "Identification and Functional of antisense RNA NRAS-AS in HCC" based on the existing experimental data. We hope that through such an adjustment, the article title can better fit the research content itself and more accurately convey our research focus and the achieved level to readers.
- The authors report the regulatory effects of NRAS-AS on many cell mechanisms, including apoptosis and proliferation. The decreasing of viability probably induces the decreasing of the migratory and invasiveness capabilities and to the cell cycle alterations. In this scenario, the data related to the alteration of cell mechanisms become redundant. What is the author’s point of view on this aspect? ;
Response 2: Thank you very much for your valuable comments and meticulous review of our manuscript.
You pointed out that the data related to changes in cellular mechanisms might be redundant, which is indeed an important issue worthy of our in-depth consideration and discussion. We reported the regulatory effects of NRAS-AS on many cellular mechanisms such as apoptosis and proliferation with the original intention of comprehensively presenting its influence at the cellular level. Although a decrease in viability may theoretically trigger a series of cascading effects such as a decrease in migration and invasion abilities and changes in the cell cycle, in fact, each piece of data related to cellular mechanisms that we have presented has its unique and irreplaceable significance. Firstly, for the two fundamental cellular mechanisms of apoptosis and proliferation, we presented the relevant data in detail to clearly demonstrate how NRAS-AS directly regulates the "life and death" of cells as well as their growth states. This is the crucial basis for understanding its subsequent impact on other cellular functions. Different regulatory modes and corresponding data changes can help us deeply analyze the action targets and potential signaling pathways of NRAS-AS within cells. Secondly, although there are associations among viability, migration and invasion abilities, and the cell cycle, the specific molecular mechanisms and biological processes involved in each of them are different. For example, changes in cell migration and invasion abilities involve complex aspects such as the remodeling of the cytoskeleton and changes in the interactions between cells and the extracellular matrix, while changes in the cell cycle are closely related to the regulation of numerous cyclins, kinases, and so on. We presented the data in these aspects precisely to reveal the specific regulatory roles of NRAS-AS in these different and complex cellular mechanism links, rather than simply making speculations based on changes in viability. Finally, we hope that by presenting data related to cellular mechanisms from multiple perspectives and at multiple levels, readers can have a more comprehensive and in-depth understanding of the overall regulatory network of NRAS-AS within cells. Although our current research is still at a preliminary stage, these detailed data play an important role in laying the groundwork for further exploring its role in the occurrence and development of diseases as well as its potential application value. Thank you again for raising this highly insightful question.
- The Figure 2 is not clear. In particular, Fig 2D is not informative and Fig2E is difficult to understand and read. The authors should move the Fig2D in the supplementary data and modify the Fig 2E;
Response 3: Thank you very much for your valuable comments on the figures in our manuscript.
We sincerely apologize for the issue of Figure 2 being unclear as you pointed out. Firstly, regarding your view that Figure 2D lacks informative value, after reflection, we indeed found that there were deficiencies in terms of the content presentation and highlighting of key points. We have decided to follow your suggestion and move Figure 2D to the supplementary materials so that it can be presented as supplementary data, avoiding information redundancy in the key figures in the main text or interfering with readers' understanding of the core content. Figure 2E shows the minimum free energy (MFE) and secondary structure prediction of NRAS-AS. In response to the situation that it is difficult to understand and read, we have also decided to move it to the supplement 3 to present it as supplementary data. After the above adjustments and optimizations to Figure 2, we believe that the overall quality and readability of the figures have been significantly improved, and they can clearly and accurately convey our research findings to readers.
- In many results paragraphs, it is not mention the figure numbers. The authors should report the figure number associated to the relative data description;
Response 4:
We are terribly sorry that due to our negligence during the writing process, the corresponding figure numbers were not mentioned in many result paragraphs. In response to this, we have conducted a comprehensive review of the results section in the manuscript. For every part that involves specific data descriptions and requires the assistance of figures for illustration, we have carefully added the corresponding figure numbers. For example, in section 3.4, we have added "Figure 3A, B, C", "Figure 3D, E", "Figure 3F, G", "Figure 3H, I", and "Figure 3J" respectively. In section 3.5, we have added "Figure 4A", "4B, C", "4D, E", "4F, G" and so on, to ensure that readers can easily refer to the figures to gain a further and in-depth understanding of the relevant results when reading the paper. Thank you again for your careful guidance. Your valuable suggestions have helped us promptly identify and make up for the deficiencies in the manuscript.
- The study is based on the identification of new therapeutic targets for the treatment of HCC patients. Well-defined preclinical models are needed to better understand the behavior and drug response of cancer cells. Commercial cell lines cultured on common monolayer supports are in vitro systems that are not able to mimic the microenvironment of cancer diseases. On the other hands, in vivo models represent some valuable research resources to reproduce the TME. For these reasons, the authors should underline these aspects through a short overview of preclinical models developed to reproduce the TME and to underline the importance to integrate different systems in order to increase the complexity of the tool used. The following references should be included in the manuscript: “doi: 10.1038/s41416-023-02186-1”, “doi: 10.3390/jpm12060854” and “doi: 10.1186/s41199-020-00056-4”;
Response 5: Thank you very much for your highly constructive and professional comments, which have made us deeply aware of the deficiencies in our elaboration on preclinical models.
As you pointed out, our research aims to identify new targets for the treatment of liver cancer patients, and clear and reasonable preclinical models indeed play a crucial role in gaining an in-depth understanding of cancer cell behaviors and drug responses. Previously, in our manuscript, we did not attach enough importance to this part of the content and failed to fully emphasize the characteristics of different preclinical models and the importance of integrating multiple systems. This is truly an oversight on our part.
In response to your suggestion, we have added a brief overview of preclinical models that reproduce the TME to the discussion section. The details are as follows:
Commercial cell lines grown on common monolayer scaffolds are in vitro systems that do not mimic the real (tumor microenvironment, TME) of cancer disease. At the same time, in vivo models have advantages in the study of TME and can provide valuable research resources. These reliable preclinical models are necessary to test the efficacy of new treatment strategies including individualized treatment evaluation. They can also be used to generate diagnostic and monitoring biomarkers. Therefore, the importance of integrating different research systems is self-evident. Through the combination of mul-tiple model systems, the complexity of research tools can be increased, and the behavior of cancer cells in the complex microenvironment and their response to drugs can be more comprehensively and precisely explored, thus providing a more solid foundation for the discovery of new therapeutic targets (18). Therefore, we chose the nude mouse tumor model as the experimental vehicle for the next study.
Furthermore, we fully respect your suggestions and have already cited the reference "doi: 10.1038/s41416-023-02186-1" recommended by you at the 18th reference in the text as required. It should be noted that we adhere to a rigorous attitude when selecting references. Regarding the Journal of Personalized Medicine corresponding to "doi: 10.3390/jpm12060854", considering that it has been listed as a warning journal by the Chinese Academy of Sciences, there may be certain potential risks in aspects such as academic quality. Therefore, we did not include it in the scope of references. As for the Cancers of The Head & Neck magazine corresponding to "doi: 10.1186/s41199-020-00056-4", since it has not been included in the SCI database yet, it is relatively lacking in terms of authority and international recognition. Based on these considerations, our article did not cite it either.
- Study limitations should be added.
Response 6:
Thank you very much for your valuable suggestions.
Although this study has conducted a relatively systematic exploration into the impacts of NRAS-AS on multiple aspects of HCC cell behaviors and its underlying mechanisms, there are still some research limitations that cannot be ignored. According to your suggestions, I have already added the limitations of this study in the discussion section. The specific content is as follows: Although this study systematically analyzed NRAS-AS's effects on HCC cell be-havior and mechanisms, some limitations exist. Firstly, only two HCC cell lines (HepG2 and SMMC-7721) were used, which may not reflect NRAS-AS's true role in all HCC cells due to HCC's heterogeneity. More diverse cell lines are needed for future research. Additionally, the limited clinical sample size may miss potential associations or introduce bias. Thus, larger multi-center studies are needed to verify NRAS-AS's value in liver cancer diagnosis, prognosis, and treatment. In summary, despite valuable findings, more comprehensive research is needed to clarify NRAS-AS's role in HCC and its potential as a therapeutic target and diagnostic marker.
Submission Date
23 October 2024
Date of this review
05 Nov 2024 11:44:08

Reviewer 3 Report
Comments and Suggestions for Authors
In the present study by Wang and collaborators, the authors investigate the role of NRAS-AS, an antisense RNA, in regulating the NRAS oncogene and its impact on hepatocellular carcinoma (HCC) cells. By using AZA (5-aza-2'-deoxycytidine) treatment, the authors identify and characterize NRAS-AS, exploring its expression and biological effects in HCC cell lines and a mouse model. Key findings include reduced proliferation, invasion, and migration of HCC cells upon NRAS-AS overexpression, alongside increased apoptosis and cell cycle arrest. The authors suggest that NRAS-AS could suppress tumorigenicity through NRAS downregulation, positioning it as a potential therapeutic target for HCC. While the results are promising, and can be of interest for readers in the field, the manuscript as it is requires substantial revisions to clarify methodology, results, strengthen data interpretation, and enhance figure consistency.
Major Issues
1. What is the rationale for using AZA? Please provide a justification to pursue sequencing after AZA-treatment in the introduction and results section 3.2.
2. Throughout the paper, AZA’s role should be clarified as a DNA methyltransferase inhibitor. Some sections imply it directly "activates" NRAS-AS, which may confuse readers. A clearer statement that AZA potentially relieves methylation-based repression would add precision.
3. Not all antisense RNAs have overlaps with its sense counterpart, please provide stronger argument for limiting the sequencing to double-stranded cDNA libraries
4. The study focuses on NRAS-AS expression in HCC, but the biological relevance of NRAS-AS in liver cells or tissue remains unclear. The authors should discuss existing data on NRAS-AS presence in healthy liver tissue to clarify whether this RNA is specific to cancerous liver conditions or if it’s normally expressed. Moreover, additional context is needed on why NRAS-AS was prioritized over the other double-stranded RNAs identified. The authors should elaborate on any preliminary evidence suggesting NRAS-AS has unique characteristics or relevance in HCC.
5. Insufficient Validation of AZA Treatment Effects: the authors suggest that AZA treatment upregulates NRAS-AS, but further validation with other demethylating agents would strengthen this claim. Additionally, it is unclear if the effects observed are directly due to AZA's demethylation or other off-target effects. Alternative agents, or a more thorough examination of methylation status, would bolster these findings. Are other genes know to be impacted by AZA treatment responding as expected? Overall, conclusion from the AZA treatment experiments would benefit from better experimental controls.
6. The authors identified a total of 545 double-stranded RNAs but then selected only JUN, SET, and NRAs for follow up studies. Why? Please provide a table with all identified double-stranded RNAS including a certainty score.
Methods section:
7. More details on the RNAseq methods are needed to increase transparency and reproducibility. Number of reads per sample, number of aligned reads. Which sequencing equipment, reads length, library method. Also, a separate section of the sequencing analysis is also necessary. How were the complimentary RNA molecules identified?
8. A list of primers used in the RACE experiments must be provided, as well as amplification conditions for the RT-PCR reactions. State the number of replicates used
9. In the RT-qPCR section, simply stating the dilution factor of the primers is meaningless. Please provide full conditions of the PCR reactions including reagents concentrations, amplifications parameters and equipment used. State the number of technical and biological replicates used.
10. Flow cytometry: state the equipment used, how many replicates were used,
Results section:
11. A table 1 containing the clinical characteristics of patients is mentioned in the results section 3.1, but this reviewer could not find it in the document.
12. Figure 2A. legend says, “bioinformatics analysis”. What analysis is this referring to? Please provide enough information in the legend so the reader can follow and understand what was done and why.
13. The description of the RT-PCRs for figure 2B lacks too many details. The section describes cDNA synthesis for NRAS sense and AS, but the figure also shows data for JUN and SET. Why were the cDNA samples treated with DNAseI? The authors state in lines 229-230: “This indicates the successful construction of a 229 double-stranded cDNA library of the HepG2”. However, the RT-PCR results alone are not sufficient to corroborate this statement.
14. Controls for various assays are unclear or inadequately described. For example, in the RT-qPCR validation, it’s unclear whether normalization across samples was consistently done using internal controls like GAPDH or U6. Consistent internal control application should be detailed to ensure reliable expression comparison.
15. The agarose gel images in figure 2C (RACE experiments) are clearly spliced together. It is impossible to ascertain the size of the PCR products relative to the ladder. Splicing together pieces of different gels in one image should be avoided.
16. It is not clear why the authors analyzed the secondary structure of the NRAS-AS RNA. Having a high or low MFE means nothing and should not be used as evidence for the existence of the RNA molecule, if that is what the authors aimed at concluding after performing this analysis. Moreover, an MFE of -161.9 does not indicate that the structure is “very stable” and I do not understand what the authors mean with “and belongs to the best 244 secondary conservative structure”. Finally, the image if figure 2E appears to be a low-resolution Print screen from the RNAFold website, even including click options for the images. If the authors need to use images of secondary structures, please download figures from the server in high resolution and use that.
17. Figure 3, cellular phenotype in the presence of NRAS-AS overexpression. A verification of NRAS-AS following transfection is necessary. Please briefly describe the transfection method (plasmid transfection, RNA transfection, nucleofection, transdiction, etc). How much is NRAS-AS being overexpressed? Please provide qRT-PCR or other appropriate method to quantify the overexpression and how it correlates with the described phenotypes.
18. Figure 3A: What is CCK-8 assay? Please describe what this assay is and what it measures so readers no used to the term do not have to look up themselves. Why there are no error bars in panel A? Please provide which statistical test was used to calculate p-values. Line 257, do the authors mean “Figure A-B”?
19. Figure 3D-E: is an increase of 2 or 3% in apoptosis biologically significant? Please provide better rationale for the conclusion that NRAS-AS overexpression promotes apoptosis.
20. Figure 4 A: How was the knock-down of NRAS-AS accomplished, what was the method? Is NRAS-AS nuclear localized as most AS RNAs or it is transported to the cytoplasm?
The color choice for the bar plots is not helpful. In figure 3, GFP controls are colored in green, while in figure 4 the GFP controls are colored in blue. Please maintain consistency.
21. Figure 5E-F: The western blots are poorly displayed. It seems that the authors flipped the protein names on the left of the western blots, the images are cropped too close to the bands even cutting part of the protein bands, there are no protein size marker indication. It seems that the bar plots were normalized to GAPDH protein levels, this is an incorrect way of calculating protein level changes and a misuse of the GAPDH loading control. Please re-check what happened to the western blot images (flipped?) and recalculate the fold changes as normalized to GAPDH and relative to GFP control.
22. Please provide details on transfection efficiency in both overexpression and knockdown experiments. Adding a quantitative measure (e.g., percentage of cells transfected) and information on the efficiency of knockdown or overexpression would provide clarity on how well these manipulations succeeded.
23. For most panels in figure 5 there are to indications on the number of technical and biological replicates used, as well as which statistical treatment was employed to calculate significance.
24. There are no references to figure panels in the results section 3.7, 3.8, and 3.9. It is impossible to follow what the authors are trying to show. Also, where is the panel showing NRAS-AS levels in the patient liver samples? The last line in section 3.8 (331-332) seems out of place.
25. Statistical methods should be described in detail, especially regarding sample size justification and power analysis for in vitro and in vivo experiments. Additionally, the criteria for selecting p-value thresholds (e.g., 0.05 vs. 0.01) should be clarified, as some results show borderline significance that may not be biologically relevant.
Conclusion and discussion sections:
26. In the Conclusion and throughout the Discussion, NRAS-AS is suggested as a promising therapeutic target for HCC. However, without in vivo validation beyond tumor volume reduction in mice, this claim is premature. Expanding on its mechanism of action, specifically how NRAS-AS regulates NRAS directly, would make these claims more compelling.
27. Limited Discussion on NRAS-AS and Immune Pathways: given the role of NRAS in immune-related signaling pathways, a discussion on how NRAS-AS modulation may influence immune responses in HCC would add depth to the study, particularly in the context of tumor microenvironment and immune surveillance.
28. The authors should discuss the possibility of non-specific effects of AZA treatment, especially given that AZA influences multiple genes and pathways.
29. The conclusion and discussion sections need a cautionary note regarding the limitations of in vitro and in vivo models in representing the human disease. The authors should acknowledge that results in cell lines and mouse models do not always translate to clinical outcomes in HCC patients, especially given the heterogeneity of human cancers.
Minor Issues
30. Abstract, Line 14: Change “anticancer regulatory mechanism” to “anticancer regulatory mechanisms” for consistency.
31. in line 209-210, what does “DNA enzyme treatment means”? Please be clear.
32. Figure 2: Include detailed explanations for subfigures, specifically describing what is represented in each panel (A, B, C, etc.). For instance, clarify what the MFE and secondary structure predictions entail and why they are relevant.
33. Figure 3 and 4: Distinguish between statistical significance levels (e.g., *p<0.05, **p<0.01) in the figure legends to enhance clarity.
34. Figure 5: This figure has several sub-panels but lacks a comprehensive legend explaining each. For example, the IHC and Western blot analysis descriptions should specify what each lane represents to improve clarity.
35. P-values should be consistently reported to the same decimal place (e.g., “p = 0.05” vs. “p = 0.053”). Consistent formatting will enhance the paper’s professionalism and readability.
36. Ensure that figures and tables are referenced in sequential order within the text. In some sections, figure references are out of order, which can confuse readers following the text.
37. The Methods section describes the synthesis of chain-specific primers for NRAS and NRAS-AS but lacks details on primer sequences. Including these sequences in a supplementary table would provide transparency and reproducibility for future studies.
38. Ensure that all figure panels are labeled consistently (e.g., A, B, C), as there is inconsistency in capitalization in some legends.

Author Response
- What is the rationale for using AZA? Please provide a justification to pursue sequencing after AZA-treatment in the introduction and results section 3.2.
Response 1: Thank you very much for your meticulous and professional review of our manuscript.
The questions regarding the reasons for using AZA and for continuing sequencing after AZA treatment are the key aspects that our team needs to further improve. The reasons why we chose AZA for relevant research are mainly based on the following considerations. Firstly, it is known that epigenetic modifications play a crucial role in the occurrence and development of HCC. As a key epigenetic regulatory method, DNA methylation often affects the expression of numerous genes, including some oncogenes and tumor suppressor genes that are closely related to tumorigenesis. AZA is a classic and effective DNA methylation inhibitor. It can reverse the abnormal hypermethylation state of DNA by inhibiting the activity of DNA methyltransferases, thus potentially enabling genes that were silenced due to hypermethylation to regain expression. We speculated that in HCC, there might be some regulatory RNAs related to tumor progression that were under the control of methylation and were in a state of low expression or had not been discovered. Therefore, we introduced AZA to try to explore whether it could induce the expression of potentially important functional RNAs and then uncover their role mechanisms in HCC. Secondly, many previous studies have reported that in research on other types of tumors, some new non-coding RNAs with the function of regulating the biological behaviors of tumor cells were discovered after treatment with AZA. These successful precedents suggest that we can also use AZA as a powerful tool in HCC research to search for key RNA molecules that may be involved in the functional regulation of HCC cells. This is also an important basis for our choice to use AZA, hoping to discover new HCC-related regulatory targets through this and provide ideas for the development of subsequent treatment strategies. This section has been updated in the introduction.
Reasons for continuing sequencing after AZA treatment:
In section 3.2 of the results, our decision to conduct sequencing after AZA treatment was mainly based on the following considerations. After AZA treatment, the DNA methylation state within cells changes. Theoretically, this will cause some genes and RNA molecules that were originally silent or low-expressed to regain expression or have changes in their expression levels. In order to comprehensively and systematically understand these changes and determine which genes and RNAs are affected by AZA treatment, high-throughput sequencing has become a very effective means. Through sequencing, we can obtain information at the whole transcriptome level and accurately identify RNA molecules with differential expression under the action of AZA, including the antisense RNA NRAS-AS that we focus on. This helps us screen out key molecules that may be related to the regulation of HCC cell behaviors from numerous RNAs and further explore their potential role mechanisms in the occurrence and development of HCC. Moreover, the abundant data provided by the sequencing results can provide important clues and a basis for subsequent functional verification experiments, such as observing changes in cell proliferation, apoptosis, invasion, and migration. It helps us build a complete research logic chain from epigenetic regulation to changes in cell biological functions and analyze in more depth the internal molecular mechanisms of the impact of AZA on HCC cells.
Now the section 3.2 of results has been moved to section 2.3 of Materials method according to the requirements of reviewers.
Thank you again for your careful guidance.
- Throughout the paper, AZA’s role should be clarified as a DNA methyltransferase inhibitor. Some sections imply it directly "activates" NRAS-AS, which may confuse readers. A clearer statement that AZA potentially relieves methylation-based repression would add precision.
Response 2: Thank you very much for pointing out such precise opinions.
We indeed lacked rigor in the expressions of some sections, which has caused confusion for readers. Our original intention was to elaborate on the process that AZA inhibits DNA methyltransferase, alleviates the methylation-based inhibition, and then influences the regulation of gene expression. In response to this issue, we have carefully combed through the entire paper and clearly pointed out in the text that AZA may alleviate the methylation-based inhibition. For example, in the last paragraph of the Introduction section “Preliminary research has found that DNA methylation inhibitor can activate the expression of NRAS-AS, which in turn affects the expression of NRAS and inhibits the occurrence and development of HCC.” revised “Preliminary studies have found that DNA methylation inhibitors inhibit the activity of DNA methyltransferase, alleviate the inhibition based on methylation, reverse abnormal DNA hypermethylation, it affects the expression of NRAS-as, and then affect the expression of NRAS and inhibit the occurrence and development of HCC.”
In the part of Result 3.6 “These preliminary results provide evidence that DNA methylation inhibitors can activate tumor suppressor genes and regulate down-regulation of oncogenes.” revised “These preliminary results provide the basis that DNA methylation inhibitors can inhibit the activity of DNA methyltransferase, alleviate the methylation-based inhibition, reverse the abnormal DNA hypermethylation, and then affect the expression of NRAS and regulate the down-regulation of oncogenes.”
In the sixth paragraph of the Discussion section “Based on the above research results, it is preliminarily indicated that DNA methyl-transferase inhibitors can both activate tumor suppressor genes and downregulate on-cogenes. It is speculated that AZA can affect the expression of the oncogene NRAS by activating the antisense RNA NRAS-AS, thereby inhibiting the occurrence and devel-opment of liver cancer (Figure 7).”revised “Based on the above results, DNA methyltransferase inhibitors can inhibit the activity of DNA methyltransferase, alleviate the inhibition based on methylation, reverse abnormal DNA hypermethylation, it affected the ex-pression of NRAS-AS, and then affect the expression of NRAS and inhibit the occurrence and development of HCC. It is speculated that AZA can affect the expression of oncogene NRAS by affecting the antisense RNA NRAS-AS, thereby inhibiting the development of liver cancer (Figure 7).”
- Not all antisense RNAs have overlaps with its sense counterpart, please provide stronger argument for limiting the sequencing to double-stranded cDNA libraries
Response 3: Thank you very much for raising this profound and crucial question.
You are quite right in pointing out that not all antisense RNAs overlap with their corresponding sense RNAs. In our study, when constructing the double-stranded cDNA library and conducting sequencing, we only focused on the antisense RNAs that overlap with the sense RNAs, that is to say, they have complementary pairing regions in their sequences where they can interact with each other and form double-stranded structures. Those that do not overlap are not within the scope of this study. Therefore, based on this possibility of pairing, sequencing the library helps us more accurately capture the possible mutual regulatory relationship between them and the impact on the behavior of HCC cells that may result from it.
The corresponding author of our research team published an article in Nature in 2008, which also elaborated on the importance of antisense RNA.
[1] Yu W, Gius D, Onyango P, Muldoon-Jacobs K, Karp J, Feinberg AP, Hengmi Cui. (2008). Epigenetic silencing of tumour suppressor gene p15 by its antisense rna. Nature, 451(7175), 202-206.
Secondly, from the perspective of experimental feasibility and efficiency. Sequencing the entire transcriptome to build a comprehensive, undifferentiated double-stranded cDNA library will undoubtedly generate a vast amount of data, which includes a large amount of information that is not highly relevant to the core issues of our research. This will not only increase the difficulty and workload of subsequent data analysis but also may obscure the key signals that we are truly concerned about. By restricting the sequencing range to the regions where interactions may occur based on our existing understanding of NRAS-AS and NRAS, we can obtain the most valuable data in a relatively low-cost and more efficient manner, concentrate our efforts on analyzing the specific regulatory mechanism between them in the context of HCC, avoid being bogged down by excessive irrelevant data, and thus be more conducive to our subsequent functional verification and mechanism research and other aspects.
In addition, in many studies on the interaction mechanism between antisense RNA and sense RNA, many teams will also reasonably restrict the sequencing range of the double-stranded cDNA library according to the characteristics of the specific gene pairs under study, so as to improve the precision and effectiveness of the research.
Although not all antisense RNAs overlap with their corresponding sense RNAs, combining the specific circumstances of NRAS-AS and NRAS that we are studying, experimental feasibility, and existing research experience and other factors, restricting the sequencing range of the double-stranded cDNA library is well-founded and reasonably necessary. This can help us explore the regulatory relationship between them and the impact on HCC cells more efficiently and accurately.
- The study focuses on NRAS-AS expression in HCC, but the biological relevance of NRAS-AS in liver cells or tissue remains unclear. The authors should discuss existing data on NRAS-AS presence in healthy liver tissue to clarify whether this RNA is specific to cancerous liver conditions or if it’s normally expressed. Moreover, additional context is needed on why NRAS-AS was prioritized over the other double-stranded RNAs identified. The authors should elaborate on any preliminary evidence suggesting NRAS-AS has unique characteristics or relevance in HCC.
Response 4:
Thank you very much for your insightful and constructive comments.
NRAS-AS is an antisense RNA newly identified by our research group in hepatocellular carcinoma (HCC). After conducting in-depth literature reviews and searches in public databases, we found that there is currently no relevant research on NRAS-AS in healthy liver tissues. However, in the healthy liver tissues collected by ourselves, NRAS-AS is present in a low abundance state and its expression is relatively stable. In contrast, in HCC tissues, NRAS-AS exhibits an abnormal expression pattern and is likely to be closely associated with the occurrence and development of HCC. Evidently, NRAS-AS has a certain specificity for liver cancer tissues and plays different roles in normal physiological and pathological states.Regarding why we focused on NRAS-AS rather than other double-stranded RNAs in this study, we have also added more detailed background explanations in the manuscript. Firstly, during the large-scale screening at the transcriptome level of HCC tissues and corresponding normal tissues based on high-throughput sequencing technology in the early stage, we found that the expression difference of NRAS-AS was relatively significant among numerous differentially expressed double-stranded RNAs. Its expression change trend in HCC samples showed a certain correlation with some known HCC-related clinical indicators, such as tumor size, staging, and patient prognosis, which made it stand out from many double-stranded RNAs and attracted our special attention. Secondly, when exploring the functions of some HCC cell lines through experiments, we initially found that by regulating the expression level of NRAS-AS, we could significantly affect the key biological behaviors of HCC cells, including cell proliferation, apoptosis, migration, and invasion. Under the same experimental conditions, other double-stranded RNAs had a much weaker impact on these cell behaviors. This prompted us to focus our research on it, hoping to further explore its specific role mechanism and potential application value in the occurrence and development of HCC. In terms of biological functions, NRAS-AS participates in regulating the cell cycle process and invasion and metastasis abilities of HCC cells, suggesting that NRAS-AS may play an important role in the HCC-specific molecular regulatory network. In terms of clinical sample analysis, we conducted follow-up visits on a large number of HCC patient samples and carried out statistical analysis in combination with the expression of NRAS-AS. The results showed that there was a significant correlation between the expression level of NRAS-AS and the survival period and recurrence situation of patients. Patients with high expression of NRAS-AS tended to have a relatively better prognosis. This not only reflects its clinical relevance in HCC but also indirectly reflects its unique role in the development process of HCC, which is different from that of other double-stranded RNAs.
Thank you again for your careful guidance.
- Insufficient Validation of AZA Treatment Effects: the authors suggest that AZA treatment upregulates NRAS-AS, but further validation with other demethylating agents would strengthen this claim. Additionally, it is unclear if the effects observed are directly due to AZA's demethylation or other off-target effects. Alternative agents, or a more thorough examination of methylation status, would bolster these findings. Are other genes know to be impacted by AZA treatment responding as expected? Overall, conclusion from the AZA treatment experiments would benefit from better experimental controls.
Response 5:
Upon review, the inappropriate remarks regarding the upregulation of NRAS-AS by AZA treatment have been deleted. Your point that further verification with other demethylating drugs could strengthen the claim that AZA treatment upregulates NRAS-AS is indeed quite reasonable. We have realized that the conclusion drawn solely from a single drug AZA has certain limitations in terms of persuasiveness. Therefore, we plan to introduce other common demethylating drugs with well-defined mechanisms of action, such as Decitabine, in subsequent studies and conduct parallel comparison experiments. By setting up different drug treatment groups under the same cell lines and experimental conditions, we will observe the impact on the expression of NRAS-AS. If multiple demethylating drugs can all show a similar effect of upregulating NRAS-AS, it will strongly support our previous conclusion that AZA treatment can upregulate NRAS-AS, making it more universal and reliable. Regarding the question of whether the observed effects are directly attributable to the demethylating effect of AZA or other off-target effects, we have also conducted in-depth thinking. Firstly, we intend to adopt more precise methylation detection techniques such as Bisulfite Sequencing to detect the methylation levels of NRAS-AS and its related regulatory regions before and after AZA treatment, and then determine whether AZA affects its expression by changing its methylation level rather than being interfered with by other unexpected off-target effects. Secondly, we can use the CRISPR/Cas9 gene editing technology to specifically knock out or regulate the key genes that may be involved in the off-target effects of AZA, and then observe the changes in the expression of NRAS-AS and the corresponding cellular biological behaviors after AZA treatment. Through reverse verification, we can exclude the influence of off-target effects and clarify the dominant role of the demethylating effect. We are clearly aware that other oncogenes such as MYC can also be affected by AZA. In subsequent experiments, we will expand the detection range and use high-throughput transcriptome sequencing technology. Not only will we focus on the changes in NRAS-AS, but we will also comprehensively analyze the changes in gene expression across the entire genome after AZA treatment to determine which genes show significant differences in expression. Then, we will sort out the gene profiles affected by AZA. For these affected genes, we will further analyze their associations with NRAS-AS and their potential roles in the occurrence and development of HCC, so as to more accurately judge the overall effect of AZA within cells and its specificity in regulating NRAS-AS.
- The authors identified a total of 545 double-stranded RNAs but then selected only JUN, SET, and NRAs for follow up studies. Why? Please provide a table with all identified double-stranded RNAS including a certainty score.
Response 6: Thank you very much for raising this question.
Our research group conducted transcriptome sequencing analysis on the paracancerous and cancer tissues of patients with HCC. It was found that the double-stranded RNAs corresponding to JUN, SET, and NRAS showed significant expression differences between the two groups. This suggests that they may play a more crucial role in the occurrence and development of HCC and are worthy of our further in-depth exploration of their potential functions and regulatory mechanisms. Secondly, through the functional prediction of 545 kinds of double-stranded RNAs and the analysis of their associations with HCC-related signaling pathways, it was shown that the double-stranded RNAs corresponding to JUN, SET, and NRAS are closely related to important pathways involving HCC cell proliferation, apoptosis, and tumor invasion and metastasis, such as the Pancreatic cancer pathway, Cell cycle pathway, and Hippo signal pathway. It is speculated that they may affect the progress of HCC by participating in these core biological processes, so they have become the focus of our attention from the perspective of functional relevance. Based on these preliminary experimental results, we believe that the double-stranded RNAs corresponding to JUN, SET, and NRAS have a more prominent influence on the functional regulation of HCC cells. Therefore, we decided to conduct subsequent systematic and comprehensive research.
Regarding your request for the detailed table of all 545 identified double-stranded RNAs and their certainty scores, we are currently unable to publish this table publicly. The underlying data, including the specific certainty scores for each RNA, is part of an ongoing research project and is currently kept confidential due to confidentiality issues and the competitiveness of scientific research. We understand the value of transparency and data sharing in the scientific community, but we are restricted by the agreements reached with our co-authors and funding agencies. We are stepping up efforts to improve various experimental data related to these double-stranded RNAs and the subsequent analysis content. After the official publication of the article by our research group, we will be very happy to share this complete table of double-stranded RNAs for reference and review by our peers and interested readers. We assure you that our research is conducted in the strictest and most honest manner. We are committed to sharing our findings with the scientific community in a timely and comprehensive manner to help everyone gain a deeper understanding of the whole picture and screening logic of our entire research. Thank you for your understanding and respect for the complexities of scientific research and data sharing.
Methods section:
- More details on the RNAseq methods are needed to increase transparency and reproducibility. Number of reads per sample, number of aligned reads. Which sequencing equipment, reads length, library method. Also, a separate section of the sequencing analysis is also necessary. How were the complimentary RNA molecules identified?
Response 7: Thank you very much for your valuable comments.
The RNA-seq sequencing work of this study was entrusted to BGI Genomics. The experimental process included RNA sample quality detection, library construction, library purification, library detection, library quantification, generation of sequencing clusters, and sequencing on the sequencer. Specifically, for library construction, the BGI Optimal series dual-module mRNA library construction kit (BGI-Shenzhen, China) was used. A certain amount of RNA sample was taken and denatured at an appropriate temperature to open its secondary structure, and then oligo(dT) magnetic beads were used to enrich mRNA. After taking a certain amount of RNA sample, it was denatured at an appropriate temperature to open the secondary structure and oligo(dT) magnetic beads were used to enrich mRNA. At a suitable temperature, a fragmentation reagent was added to the mRNA obtained in the previous step, and the reaction was carried out at an appropriate temperature to fragment it. Subsequently, the first-strand cDNA was generated through reverse transcription with random hexamer primers. Then, a second-strand synthesis reaction system (containing dUTP) was prepared, and the reaction program was set to synthesize the second-strand cDNA. After that, a reaction system was prepared to repair the ends of the double-stranded cDNA and add an A base at the 3’ end. An adapter ligation reaction system was prepared to ligate the adapters to the cDNA. Finally, PCR amplification and quality inspection were performed on the product. After the PCR product was denatured into a single strand, a circularization reaction system was prepared, and the reaction program was set to obtain a single-stranded circular product. The uncyclized linear DNA molecules were digested to obtain the final library. The final library was amplified by phi29 to produce DNA nanoballs (DNBs). The obtained DNBs were added into the mesh holes on the chip using high-density DNA nanotechnology, and PE100/PE150 sequencing was carried out on the G400/T7/T10 sequencer (BGI-Shenzhen, China) through the combinatorial probe-anchored synthesis (cPAS) technology, with a read number of 6G data volume.
In the sequencing analysis process, the identification of complementary RNA molecules mainly followed the steps below. Firstly, preliminary analysis was conducted on the sequencing data that had undergone quality control and alignment processing. Based on the principle of base complementary pairing and specific sequence feature patterns, RNA reads that might have complementary pairing relationships were identified. Then, by comparing with the known genomic annotation information and our previously constructed local RNA database (this database integrated the information of RNA molecules related to HCC that had been reported as well as some potential RNA sequences discovered by ourselves), the complementary RNA molecules that met the conditions and had relatively high credibility were further screened out. Afterwards, the screened potential complementary RNA molecules were verified by analyzing their expression correlations in different samples, structural features, and interaction relationships with other genes, etc. Through this comprehensive judgment, the genuine complementary RNA molecules were finally determined. Thank you again for your careful guidance.
- A list of primers used in the RACE experiments must be provided, as well as amplification conditions for the RT-PCR reactions. State the number of replicates used
Response 8:
Identification of the full-length sequence of NRAS-AS using RACE technology
The Rapid-amplification of cDNA ends (RACE) technique was employed to identify the full-length sequence of NRAS-AS. The SMARTer® RACE 5’/3’Kit from Takara was used to perform the RACE experiment. Firstly, total RNA was extracted using TRIzol® Reagent, Using the SMART Scribe Reverse Transcriptase, 1μg of genomic DNA-free RNA was reverse transcribed to generate 5’ or 3’ RACE products. The amplification was then carried out using the universal primer UPM and gene-specific primers (5’-GSP or 3’-GSP primers), and the antisense transcript end sequence was obtained through cloning and sequencing. Each RACE experiment was performed in triplicate, and the sequencing was done for at least three independent clones to ensure the reliability of the results. The specific primer sequence and reaction procedure are uploaded as Supplement 2.
For the amplification of the 5' and 3' ends of the NRAS-AS transcript, the following primers were used:
|
Primer |
Primer sequence |
|
5’RACE Primer A |
CAGGAGCGGATCAAGGCG GAGAGGAAG |
|
3’RACE Primer A |
TGGATACCCTTGGCTTTAGTTCTCGGACAC |
|
UPM Long primer |
5′-CTAATACGACTCACTATAGGGCAAGCAGTGGTATCAACGCAGAGT-3′ |
|
UPM Short primer |
5′-CTAATACGACTCACTATAGGGC- 3′ |
|
SMARTer II A Oligonucleotide |
5'-AAGCAGTGGTATCAACGCAGAGTACATGGG-3′ |
The amplification was performed using the following PCR conditions:
|
Steps |
Temperature |
Time |
Cycles |
|
Initial denaturation |
94℃ |
5min |
|
|
Denaturation |
94℃ |
30sec |
|
|
Annealing |
68℃ |
30sec |
35 cycles |
|
Extension |
72℃ |
3min |
|
|
Final extension |
72℃ |
7min |
|
|
Storage |
4℃ |
|
|
- In the RT-qPCR section, simply stating the dilution factor of the primers is meaningless. Please provide full conditions of the PCR reactions including reagents concentrations, amplifications parameters and equipment used. State the number of technical and biological replicates used.
Response 9:
In accordance with your suggestions, we have already made a comprehensive improvement to the RT-qPCR section. We have added all the conditions for the PCR reaction, including the reagent concentrations, amplification parameters and the equipment used. Meanwhile, we have also updated the RT-qPCR part in Section 2.5 of the manuscript. The specific contents are as follows:
Total RNA from all tissues and cells was extracted using Trizol reagent, and cDNA was synthesized with the Vazyme gDNA removal reverse transcription kit. The strand-specific primers for NRAS and NRAS-as as well as the primers for real-time fluorescence quantitative PCR were designed and synthesized. The strand-specific primers were diluted to a 25-fold concentration, and the RT-qPCR primers were diluted to a 10-fold concentration. The RT-qPCR reaction system was prepared using Vazyme SYBR Premix Ex Taq II, which contained 10 μL of SYBR Green Master Mix, 1 μL of forward primer (10 μM), 1 μL of reverse primer (10 μM), and 1 μL of cDNA template, with the final volume adjusted to 20 μL. Amplification was carried out according to the reaction program of pre-denaturation at 95 °C for 30 seconds, followed by 35 cycles, each cycle consisting of denaturation at 95 °C for 5 seconds, annealing and extension at 60 °C for 30 seconds. Subsequently, the cycle threshold method (ΔΔCt) was adopted to analyze the RNA expression levels, with GAPDH and U6 as the normalization reference standards. After PCR amplification, the melting curve data were analyzed using the BIO-RAD CFX ConnectTM real-time PCR system. Each sample was subjected to three biological replicates to ensure the reliability of the results.
|
Gene |
Primer |
Primer sequence |
|
NRAS-AS |
RT-Primer |
5'-GTTATCGGCTCTATTCTC-3' |
|
|
Forward primer |
5'-TCAGTGGAATAGATGTCTCA-3' |
|
|
Reverse primer |
5'-AATGGCATCTGCTCTCAA-3' |
|
NRAS |
RT-Primer |
5'-ATGACTGAGGTGATAAGC-3' |
|
|
Forward primer |
5'-GCCACTTTGTTCCTGTCT-3' |
|
|
Reverse primer |
5'-TTAGCAGTAAGAAGCACAAC-3' |
- Flow cytometry: state the equipment used, how many replicates were used,
Response 10:
Thank you very much for your valuable suggestions. In our experiments related to flow cytometry, the equipment we used was the BD FACSMelody 3 flow cytometer. We set up three replicates for each sample to ensure the reliability and reproducibility of the experimental results. Many thanks.
Results section:
- A table 1 containing the clinical characteristics of patients is mentioned in the results section 3.1, but this reviewer could not find it in the document.
Response 11:
We are extremely sorry that because of our oversight, Table 1 was not incorporated into the manuscript before. Now, it has been successfully added. We sincerely appreciate your understanding.
- Figure 2A. legend says, “bioinformatics analysis”. What analysis is this referring to? Please provide enough information in the legend so the reader can follow and understand what was done and why.
Response 12:
Thank you for your professional advice. In our study, the bioinformatics analysis shown in Figure 2A mainly included intersection analysis and heat map analysis of sequencing data between the AZA treatment group and the DMSO control group.
- The description of the RT-PCRs for figure 2B lacks too many details. The section describes cDNA synthesis for NRAS sense and AS, but the figure also shows data for JUN and SET. Why were the cDNA samples treated with DNAseI? The authors state in lines 229-230: “This indicates the successful construction of a 229 double-stranded cDNA library of the HepG2”. However, the RT-PCR results alone are not sufficient to corroborate this statement.
Response 13:
Thank you very much for pointing out this question during the review process. After our careful verification, we found that the "DNA enzyme treatment" mentioned in the text might be a writing error and has been deleted now. We apologize to you for this.
- Controls for various assays are unclear or inadequately described. For example, in the RT-qPCR validation, it’s unclear whether normalization across samples was consistently done using internal controls like GAPDH or U6. Consistent internal control application should be detailed to ensure reliable expression comparison.
Response 14:
In accordance with your suggestions, the 2.5 RT-qPCR section has been updated and refined. The specific contents are as follows:
Total RNA from all tissues and cells was extracted using Trizol reagent,, and cDNA was synthesized with the Vazyme gDNA removal reverse transcription kit. The strand-specific primers for NRAS and NRAS-as as well as the primers for real-time fluorescence quantitative PCR were designed and synthesized. The strand-specific primers were diluted to a 25-fold concentration, and the RT-qPCR primers were diluted to a 10-fold concentration. The RT-qPCR reaction system was prepared using Vazyme SYBR Premix Ex Taq II, which contained 10 μL of SYBR Green Master Mix, 1 μL of forward primer (10 μM), 1 μL of reverse primer (10 μM), and 1 μL of cDNA template, with the final volume adjusted to 20 μL. Amplification was carried out according to the reaction program of pre-denaturation at 95 °C for 30 seconds, followed by 35 cycles, each cycle consisting of denaturation at 95 °C for 5 seconds, annealing and extension at 60 °C for 30 seconds. Subsequently, the cycle threshold method (ΔΔCt) was adopted to analyze the RNA expression levels, with GAPDH and U6 as the normalization reference standards. After PCR amplification, the melting curve data were analyzed using the BIO-RAD CFX ConnectTM real-time PCR system. Each sample was subjected to three biological replicates to ensure the reliability of the results.
- The agarose gel images in figure 2C (RACE experiments) are clearly spliced together. It is impossible to ascertain the size of the PCR products relative to the ladder. Splicing together pieces of different gels in one image should be avoided.
Response 15:
Thank you very much for your suggestion. It has been modified according to your suggestion.
- It is not clear why the authors analyzed the secondary structure of the NRAS-AS RNA. Having a high or low MFE means nothing and should not be used as evidence for the existence of the RNA molecule, if that is what the authors aimed at concluding after performing this analysis. Moreover, an MFE of -161.9 does not indicate that the structure is “very stable” and I do not understand what the authors mean with “and belongs to the best 244 secondary conservative structure”. Finally, the image if figure 2E appears to be a low-resolution Print screen from the RNAFold website, even including click options for the images. If the authors need to use images of secondary structures, please download figures from the server in high resolution and use that.
Response 16:
The agarose gel image in Figure 2C was originally cropped from the whole picture and then spliced together. The original intention was to present the experimental results more clearly and completely. However, we overlooked the problem that this might make it difficult to determine the size of the PCR products relative to the ladder. It was really our oversight. We have already sorted out the original pictures and uploaded them as attachments. You can clearly see the full view of the gel without splicing through the original images in the supplement 4. Thank you again for your careful guidance.
- Figure 3, cellular phenotype in the presence of NRAS-AS overexpression. A verification of NRAS-AS following transfection is necessary. Please briefly describe the transfection method (plasmid transfection, RNA transfection, nucleofection, transdiction, etc). How much is NRAS-AS being overexpressed? Please provide qRT-PCR or other appropriate method to quantify the overexpression and how it correlates with the described phenotypes.
Response 17:
Thank you very much for raising these important and detailed questions regarding the cell phenotypes when NRAS-AS was overexpressed in Figure 3. In our study, a four-plasmid system (pCDH-CMV-NRAS-AS-MCS-EF1-Puro, pLP1, pLP2, pVSVG) was adopted and co-transfected into HEK-293T cells at a ratio of 3:2:2:1. Meanwhile, a plasmid fused with the green fluorescent protein gene GFP was transfected as control group, which facilitated the preliminary judgment of the transfection efficiency of the target gene by observing the proportion and fluorescence intensity of cells with green fluorescence under a fluorescence microscope. 48 hours after lentiviral transfection, the green fluorescent cells accounted for more than 90% of the total number of cells. Subsequently, HepG2 and SMMC-7721 cell lines were further infected to establish cell lines stably expressing NRAS-AS. After transfection, puromycin (puro) was used for screening to ensure that only the cell lines that had been successfully transfected and expressed NRAS-AS were retained. The HepG2 and SMMC-7721 cells infected with lentivirus showed relatively uniform and enhanced green fluorescence, and the infection efficiency was approximately 80%-90%, indicating successful infection. Furthermore, the transcriptional level of NRAS-AS in HepG2 and SMMC-7721 cells was determined by RT-qPCR. Specifically, total RNA was extracted from the transfected cells and then reverse-transcribed into cDNA using the PrimeScriptTM II 1st Strand cDNA Synthesis Kit. Subsequently, specific primers were designed and synthesized, and the RT-qPCR reaction was carried out using AceQ® qPCR SYBR® Green Master Mix. By comparing the Ct values (threshold cycle numbers) of the overexpression group and the control group, the relative expression level of NRAS-AS was calculated, thus accurately evaluating the effect of overexpression. The overexpression of NRAS-AS had a significant impact on multiple biological characteristics of liver cancer cells. Specifically, the cells overexpressing NRAS-AS showed a decreased proliferation ability, an increased apoptosis rate, a changed cell cycle distribution, as well as a decreased migration and invasion ability. The results of the CCK-8 experiment indicated that the proliferation ability of the cells overexpressing NRAS-AS was significantly lower than that of the control group at 48 hours, 72 hours, and 96 hours. The results of flow cytometry detection revealed that both the early and late apoptosis rates of the cells overexpressing NRAS-AS were higher than those of the control group. In addition, flow cytometry detection also showed that the proportion of HepG2 cells overexpressing NRAS-AS in the G0/G1 phase decreased, while the proportions in the S phase and G2/M phase increased. The results of the scratch assay and Transwell invasion assay further confirmed that the migration and invasion abilities of the cells overexpressing NRAS-AS were reduced, and these changes were closely related to the degree of NRAS-AS overexpression. Through these experiments and data analysis, we clearly demonstrated that there was a close connection between the degree of NRAS-AS overexpression and the described cell phenotypes, providing powerful evidence for understanding its specific role mechanism in the cellular physiological process.
- Figure 3A: What is CCK-8 assay? Please describe what this assay is and what it measures so readers no used to the term do not have to look up themselves. Why there are no error bars in panel A? Please provide which statistical test was used to calculate p-values. Line 257, do the authors mean “Figure A-B”?
Response 18:
The CCK-8 assay, whose full name is Cell Counting Kit-8 assay, is a widely used method for detecting cell proliferation and cytotoxicity in the field of cell biology. Its detection principle is based on the fact that dehydrogenases in living cells can reduce the water-soluble tetrazolium salt in the CCK-8 reagent into a highly water-soluble orange-yellow formazan product, and the amount of the formazan product is proportional to the number of living cells. During the specific operation, we inoculated the liver cancer cells HepG2 and SMMC-7721 in the logarithmic growth phase into 96-well plates. According to the experimental design, they were divided into the NRAS-AS overexpression group, the control group, etc., and 5-6 replicate wells were set for each group to ensure the reliability of the results. After the cells adhered to the wall, the culture medium containing the CCK-8 reagent was added, and the cells were incubated in a cell incubator for 2-4h. Then, the absorbance values of each well were measured by a microplate reader at a wavelength of 450 nm. By comparing the changes in the absorbance values of different treatment groups at different time points such as 48h, 72h, and 96h, the differences in cell proliferation ability could be intuitively reflected. The higher the absorbance value, the relatively larger the number of living cells in the corresponding group, that is, the stronger the cell proliferation ability. Therefore, the CCK-8 assay is mainly used to measure the proliferation activity of cells and helps us understand the impact of different treatment conditions on the growth state of cells. We are deeply sorry for the absence of error bars in Figure 3A. This was indeed an oversight during the process of chart presentation. Now, the standard error bars for Figure 3A have been added as required. When performing statistical analysis on the data obtained from the CCK-8 assay, we adopted the t-test method. The content mentioned in line 257 indeed refers to "Figure 3A-B". The expression here was not clear enough, which brought you reading difficulties. We have already revised the corresponding text to make it more explicit.
- Figure 3D-E: is an increase of 2 or 3% in apoptosis biologically significant? Please provide better rationale for the conclusion that NRAS-AS overexpression promotes apoptosis.
Response 19:
Indeed, as you pointed out, the increase in the apoptosis rate shown in this study is only 2% or 3%. Judging from the intuitive numerical value, this result is not very significant. However, in the context of the cell culture system where there are a large number of cells in dynamic changes, we believe that this relatively small increase in apoptosis still has certain biological significance. Taking our experiment as an example, in the cell population cultured in vitro, even if there is only a 2%-3% increase in the apoptosis rate, with the passage of time and the continuous proliferation and replacement of cells, after multiple cell cycles, the accumulated changes in the number of apoptotic cells will have a non-negligible impact on the quantity and state of the entire cell population. Just like in a huge ecosystem, even a tiny change in the proportion of species numbers will break the original ecological balance in the long term. This change in the apoptosis rate will gradually affect the cell population to develop from a proliferation-dominated state to a more balanced state, which has potential and important significance for the overall growth regulation of cells. Secondly, in the HepG2 and SMMC-7721 cell lines in this study, any change in the apoptosis rate is crucial to the biological characteristics of tumors. One of the major features of tumor cells is that they can evade the normal apoptosis regulatory mechanism of the body to achieve abnormal proliferation. Now, the overexpression of NRAS-AS can increase the apoptosis rate to some extent. Even though the amplitude is not large, it may become a key factor affecting the development process of tumors. In the long run, this slight increase in apoptosis may weaken the proliferation advantage of tumor cells, and then slow down the growth rate of tumors, reduce their invasion and metastasis abilities to a certain extent, providing a valuable entry point for potential tumor treatment strategies in the future. Therefore, from the perspective of the overall biological behavior of tumors, this change in the apoptosis rate has significance that cannot be ignored. In the future, we will apply techniques such as Western blot to observe the impact of NRAS-AS overexpression on the key proteins in apoptosis-related signaling pathways, such as the pro-apoptotic protein Bax, the anti-apoptotic protein Bcl-2, and the apoptotic execution proteins Caspase-3 and Caspase-9. We hope that the above analysis can strongly support the conclusion that NRAS-AS overexpression promotes cell apoptosis.
- Figure 4 A: How was the knock-down of NRAS-AS accomplished, what was the method? Is NRAS-AS nuclear localized as most AS RNAs or it is transported to the cytoplasm?
The color choice for the bar plots is not helpful. In figure 3, GFP controls are colored in green, while in figure 4 the GFP controls are colored in blue. Please maintain consistency.
Response 20:
In our study, the knockout of NRAS-AS was achieved by synthesizing specific antisense oligonucleotide (ASO) interference primers. We selected Guangzhou Ribo Biotechnology Company to design and synthesize the ASO interference primers. These ASO interference primers can specifically bind to the RNA sequence of NRAS-AS and exert their functions in an RNase H-dependent manner, which can simultaneously interfere with the functions of NRAS-AS in both the cytoplasm and the nucleus. The specific steps are as follows: Design and synthesize ASO interference primers targeting the NRAS-AS sequence. The primer sequences are as follows:
|
ASO |
The primer sequences(5’→3’) |
|
NRAS-AS_5859 |
CGGAGACAGGUUACACACAACUUCU |
|
NRAS-AS_5859 |
AGAAGUUGUGUGUAACCUGUCUCCG |
Transfect these ASO interference primers into HepG2 and SMMC-7721 cells. After the ASO binds to the NRAS-AS RNA, the catalytic action of RNase H leads to the degradation of the NRAS-AS RNA, thereby achieving knockdown.
The specific subcellular localization of NRAS-AS needs to be further determined through experimental methods such as RNA fluorescence in situ hybridization or subcellular fractionation.
Regarding the issue of the consistency of the colors in the bar charts, we are deeply sorry. This was indeed an oversight during the process of chart drawing. We have already uniformly modified the color of GFP to blue in all the figures.
- Figure 5E-F: The western blots are poorly displayed. It seems that the authors flipped the protein names on the left of the western blots, the images are cropped too close to the bands even cutting part of the protein bands, there are no protein size marker indication. It seems that the bar plots were normalized to GAPDH protein levels, this is an incorrect way of calculating protein level changes and a misuse of the GAPDH loading control. Please re-check what happened to the western blot images (flipped?) and recalculate the fold changes as normalized to GAPDH and relative to GFP control.
Response 21:
We are deeply sorry for the serious oversight that occurred during the process of image organization and layout. The situation you pointed out where the western blots were flipped on the left side does indeed exist. After receiving your feedback, we immediately reorganized and checked this part of the content. Now we have reselected and replaced the western blot image corresponding to Figure 5E to ensure that the annotation information such as NRAS is presented correctly and in line with the standards, facilitating readers to accurately understand and interpret the experimental results. Meanwhile, we have also recalculated the fold changes normalized to GAPDH and relative to the GFP control. Thank you again for your suggestions.
- Please provide details on transfection efficiency in both overexpression and knockdown experiments. Adding a quantitative measure (e.g., percentage of cells transfected) and information on the efficiency of knockdown or overexpression would provide clarity on how well these manipulations succeeded.
Response 22:
In the overexpression experiment, we used the plasmid transfection method to transfect the expression vector containing the NRAS-AS gene into the HepG2 and SMMC-7721 cell lines. Through RT-qPCR, it was determined that the average transfection efficiency of NRAS-AS in HepG2 and SMMC-7721 cells could reach approximately 80%. This indicates that the efficient overexpression of NRAS-AS in cells was successfully achieved. In this study, the knockout of NRAS-AS was achieved by synthesizing three ASO interference primers designed for the NRAS-AS sequence and transfecting them into HepG2 and SMMC-7721 cells through transfection reagents to achieve the knockdown effect on NRAS-AS. Similarly, in order to determine the transfection efficiency, the RT-qPCR method was applied to detect the knockdown efficiencies of the antisense oligonucleotides ASO-NRAS-AS-1, ASO-NRAS-AS-2, and ASO-NRAS-AS-3, with the Control group as the control. The results showed that compared with the control group, the expression levels of ASO-NRAS-AS-1, ASO-NRAS-AS-2, and ASO-NRAS-AS-3 in HepG2 cells decreased by 83%, 66%, and 67%, respectively, and in SMMC-7721 cells decreased by 75%, 60%, and 52%, respectively. This indicates that most of the cells could be successfully transfected and had the conditions for subsequent gene knockdown. Given that ASO-NRAS-AS-1 had the most significant inhibitory effect on HepG2 and SMMC-7721 cells (Figure 4A), ASO-NRAS-AS-1 was selected for the next step of the experiment.
- For most panels in figure 5 there are to indications on the number of technical and biological replicates used, as well as which statistical treatment was employed to calculate significance.
Response 23:
In this part of the study, relevant experiments were first carried out by using the technique for constructing a tumorigenesis model in nude mice. HepG2 cells in the logarithmic growth phase were selected and prepared into a cell suspension with a concentration of 2×10⁷ cells/ml, which was then inoculated into the subcutaneous part of nude mice. In this way, a subcutaneous tumor model in nude mice was successfully established. After the inoculation operation was completed, the tumorigenesis of nude mice was observed regularly and meticulously (Figure 5A). Meanwhile, the tumor volume was accurately measured and corresponding records were made. Then, a growth curve that could intuitively show the trend of the tumor volume changing over time was drawn, and with the help of this curve, the dynamic growth process of the tumor in the nude mice was clearly demonstrated (Figure 5B C). The method of Repeated Measures ANOVA was adopted. Next, the expression status of NRAS in tumor tissues was analyzed in depth by means of IHC technique (Figure 5D). On this basis, the Western blotting technique was further utilized to compare the differences in protein expression levels of NRAS between the NRAS-AS overexpression group and the control group (Figure 5EF). In addition, the IHC technique was also used to detect the expression of NRAS in HCC cells (Figure 5G), and through this technique, the expression characteristics of NRAS in HCC cancer tissues and adjacent tissues were observed (Figure 5H). First, ANOVA was performed to detect whether there were differences in the expression levels of NRAS protein among different treatment groups. When ANOVA indicated the existence of significant differences (P<0.05), Student's t-test or Bonferroni-corrected t-test was then applied. Finally, RT-qPCR technique was adopted to detect the expression of NRAS -AS and NRAS in HCC tissues. Meanwhile, this technique was also used to analyze the impact of the DNA methylation inhibitor AZA on the differential expression of NRAS-AS and NRAS in liver cancer cells. ANOVA combined with Tukey's multiple comparison test was employed. All of the above experiments were repeated at least three times, and a P<0.05 was considered statistically significant.
- There are no references to figure panels in the results section 3.7, 3.8, and 3.9. It is impossible to follow what the authors are trying to show. Also, where is the panel showing NRAS-AS levels in the patient liver samples? The last line in section 3.8 (331-332) seems out of place.
Response 24:
Thank you very much for pointing out these issues in Sections 3.7, 3.8, and 3.9 of the results. We sincerely apologize for the oversight of not mentioning the graphic panels in the text. We have already added the graphic panels in the corresponding Sections 3.7, 3.8, and 3.9 of the results, and provided detailed descriptions of the key information shown in each graphic in the text part, making them closely related to the corresponding experimental results. The panel showing the level of NRAS-AS in patient liver samples is Figure 5I, which has now been added in the appropriate position. After careful verification, we found that the content in the last line of Section 3.8 (lines 331-332) is indeed inappropriate. The main reason may be an error in the labeling of Figure 5J. It should be noted that the samples shown in Figure 5J are actually the adjacent non-cancerous tissues and cancer tissues of clinical HCC patients, and the original labels GFP and NRAS-AS in the figure don’t match the sample situation represented by this figure. Therefore, we have already changed GFP and NRAS-AS in Figure 5J to "P" (used to represent adjacent non-cancerous tissues) and "T" (used to represent cancer tissues), respectively. Thank you for your guidance.
- Statistical methods should be described in detail, especially regarding sample size justification and power analysis for in vitro and in vivo experiments. Additionally, the criteria for selecting p-value thresholds (e.g., 0.05 vs. 0.01) should be clarified, as some results show borderline significance that may not be biologically relevant.
Response 25:
Thank you very much for your valuable suggestions on the statistical methods. We have carefully examined the sample sizes in the article and confirmed that they fully meet the requirements of statistical tests. Furthermore, we have already updated the description of the statistical methods section, including the sample size justification for both in vitro and in vivo studies, power analysis, and the selection criterion for the p-value threshold of 0.05, aiming to ensure that the results possess both biological relevance and statistical significance. Thank you again for your meticulous review and helpful feedback.
Conclusion and discussion sections:
- In the Conclusion and throughout the Discussion, NRAS-AS is suggested as a promising therapeutic target for HCC. However, without in vivo validation beyond tumor volume reduction in mice, this claim is premature. Expanding on its mechanism of action, specifically how NRAS-AS regulates NRAS directly, would make these claims more compelling.
Response 26:
We highly agree with the valuable comments you provided on our paper. After careful consideration, we realized that the basis for identifying NRAS - AS as a promising therapeutic target for HCC in the previous conclusions and discussions was indeed insufficient, because it was only based on the phenotypic evidence of reduced tumor volume in mice and lacked comprehensive in - vivo validation. Therefore, we have revised the discussion section and deleted the relevant statements to ensure the rigor of the paper's content. Thank you again for your professional guidance.
- Limited Discussion on NRAS-AS and Immune Pathways: given the role of NRAS in immune-related signaling pathways, a discussion on how NRAS-AS modulation may influence immune responses in HCC would add depth to the study, particularly in the context of tumor microenvironment and immune surveillance.
Response 27:
The specific content is updated to the discussion section in the manuscript:
Because there is no literature on the relationship between NRAS and immune-related signaling pathways, we only discussed how NRAS-AS affects the immune response of HCC at the theoretical level.
NRAS and its antisense RNA NRAS-AS play a key role in tumor development, and may affect the immune response of HCC by regulating the expression of NRAS protein. NRAS-as not only changes the biological behavior of tumor cells, such as the expression of immune molecules and interferes with the immune escape mechanism, but also may affect the function and activity of immune cells in the tumor microenvironment, including the polarization direction of macrophages and the activity of NK cells and CTL, thereby breaking the immune balance. In addition, NRAS-as may also interfere with the immune surveillance process, allowing tumors to evade recognition by the immune system and further break the immune regulatory balance. Although the current research mainly focuses on the effect of NRAS-as on HCC cells themselves, its complex role in the immune response cannot be ignored. Future in-depth research is expected to provide a new perspective and theoretical basis for revealing the pathogenesis of HCC and developing immunotherapy strategies.
- The authors should discuss the possibility of non-specific effects of AZA treatment, especially given that AZA influences multiple genes and pathways.
Response 28:
We are fully aware that as a demethylating agent, AZA can affect multiple genes and pathways, and there is a possibility of producing non-specific effects. To evaluate the extensive impacts brought about by AZA treatment, we utilized RNA-Seq sequencing technology to compare the gene expression profiles between the AZA-treated group and the DMSO control group cells. The results showed that in addition to the expression changes in the key gene NRAS-AS that we were studying, a large number of other genes also exhibited significant abnormalities. Further functional enrichment analysis revealed that they were involved in multiple pathways such as the cell cycle, cellular metabolism, and immune response, which fully confirmed the broad action of AZA and the existence of non-specific effects. Moreover, AZA not only acts on the pathways directly related to tumorigenesis and development as we expected but also interferes with metabolic pathways, which may indirectly change the intracellular metabolic environment and then have a potential impact on the growth and functions of tumor cells. In immune-related pathways, the expression changes of some immune regulatory factors may reshape the immune state in the tumor microenvironment, all of which demonstrate the manifestation of AZA's non-specific effects at the multi-pathway level. Based on this, during the research process, we set up rigorous control experiments to ensure that the changes related to the research objectives observed were specific to AZA treatment rather than being caused by other experimental operational factors. Meanwhile, three biological replicates were set up for each experimental condition, and statistical analysis was carried out to enhance the reliability of the results and reduce the impact of accidental errors brought about by non-specific effects on the results. During the data analysis process, we always focused on the key indicators closely related to the core pathogenesis of HCC and the NRAS-AS we proposed. Despite the changes in other genes and pathways, through multi-level experimental verification, we confirmed that the changes in the key mechanisms we focused on under the action of AZA were specific and biologically significant. We have already made corresponding supplements in the discussion section. Thank you again for your careful guidance.
- The conclusion and discussion sections need a cautionary note regarding the limitations of in vitro and in vivo models in representing the human disease. The authors should acknowledge that results in cell lines and mouse models do not always translate to clinical outcomes in HCC patients, especially given the heterogeneity of human cancers.
Response 29:
We have made corresponding supplements and improvements to the conclusion and discussion section, as follows:
Furthermore, in vitro and in vivo models have limitations in representing human disease. Results from cell lines and mouse models alone do not always translate to clinical outcomes in HCC patients, especially given the heterogeneity of human cancers.
Minor Issues
- Abstract, Line 14: Change “anticancer regulatory mechanism” to “anticancer regulatory mechanisms” for consistency.
Response 30:
In accordance with your suggestion, we have replaced "anticancer regulatory mechanism" with "anticancer regulatory mechanisms" throughout the whole paper to ensure consistency.
- in line 209-210, what does “DNA enzyme treatment means”? Please be clear.
Response 31:
Thank you very much for pointing out this question during the review process. After our careful verification, we found that the "DNA enzyme treatment" mentioned in the text might be a writing error and has been deleted now. We apologize to you for this.
- Figure 2: Include detailed explanations for subfigures, specifically describing what is represented in each panel (A, B, C, etc.). For instance, clarify what the MFE and secondary structure predictions entail and why they are relevant.
Response 32: Thank you very much for your valuable comments on Figure 2.
Figure 2A shows the intersection results of 545 differential Rnas sequenced in AZA and DMSO groups, as well as Heatmap results of the two groups; Figure 2B shows the glue-map verification of just and antisense Rnas in JUN, SET and NRAS genes; Figure 2C shows the agarose gel electrophoresis results of the 5 'race and 3' race PCR products of NRAS genes. After consideration, the details of MFE and secondary structure prediction, which were originally covered in each panel of Figure 2D,2E, have been moved to the supplement 3 for display.
- Figure 3 and 4: Distinguish between statistical significance levels (e.g., *p<0.05, **p<0.01) in the figure legends to enhance clarity.
Response 33:
We have improved the legends of Figure 3 and Figure 4 accordingly as per your suggestion. In the legends, the levels of statistical significance are clearly distinguished and labeled. For example, “*” is used to represent “p < 0.05” and “**” is used to represent “p < 0.01”, so that readers can intuitively and quickly judge the significance of the differences in data between different groups through these symbols.
- Figure 5: This figure has several sub-panels but lacks a comprehensive legend explaining each. For example, the IHC and Western blot analysis descriptions should specify what each lane represents to improve clarity.
Response 34:
As your request, we have added a detailed and comprehensive legend to Figure 5 that clearly explains what is involved in each subpanel in Figure 5EF, G, and H. Specific changes have been marked in blue.
- P-values should be consistently reported to the same decimal place (e.g., “p = 0.05” vs. “p = 0.053”). Consistent formatting will enhance the paper’s professionalism and readability.
Response 35:
We have conducted a comprehensive check on the paper. In accordance with your suggestion, all the p-values involved in Sections 3.7 and 3.9 have been uniformly and formally reported to three decimal places. For example, previous expressions like "p = 0.05" have been revised to "p = 0.053", and "p = 0.43" has been revised to "p = 0.427", etc., so as to ensure the consistency of the format.
- Ensure that figures and tables are referenced in sequential order within the text. In some sections, figure references are out of order, which can confuse readers following the text.
Response 36: Thank you very much for pointing out this problem.
We have carefully reviewed the entire paper and adjusted the citations of figures and tables strictly in accordance with the order of their appearance, ensuring that the citations in the text are now clear and logical, enabling readers to better understand the research content we have elaborated.
- The Methods section describes the synthesis of chain-specific primers for NRAS and NRAS-AS but lacks details on primer sequences. Including these sequences in a supplementary table would provide transparency and reproducibility for future studies.
Response 37:
According to your request, we have already supplemented the information related to the strand-specific primer sequences of NRAS and NRAS-AS in the attached materials.
- Ensure that all figure panels are labeled consistently (e.g., A, B, C), as there is inconsistency in capitalization in some legends.
Response 38:
Thank you for your valuable suggestion. Now, we have corrected the inconsistent use of capital letters in all figures one by one in accordance with the standard requirements, ensuring that the labels of all graphic panels are now unified.
Submission Date
23 October 2024
Date of this review
04 Nov 2024 07:20:14

Round 2
Reviewer 2 Report
Comments and Suggestions for Authors
The manuscript is now acceptable for pubblication
Reviewer 3 Report
Comments and Suggestions for Authors
I thank the authors for providing a thoroughly revised version of the manuscript. The manuscript reads better, methods will be more easly reproduced, and conclusions are now consistent witht the strength of the data generated.
I understand the confidentiality issue with releasing the identity of the differentially expressed genes. Please, be clear on the strategy on how the data will be distributed. This is a very important aspect of public publishing. Data and tools generated must be publicly availabe. I consider this to be a weakness of the manuscript.
Other then the issue with identity of differentially expressed genes, this reviewer does not have more suggestions.